Resource

# Profound phenotypic and epigenetic heterogeneity of the HIV-1-infected CD4+ T cell reservoir

Vincent H. Wu [1,2], Jayme M. L. Nordin[1,2], Son Nguyen [1,8], Jaimy Joy[2,3], Felicity Mampe[2,3], Perla M. del Rio Estrada[4], Fernanda Torres-Ruiz [4], Mauricio González-Navarro[4], Yara Andrea Luna-Villalobos[4], Santiago Ávila-Ríos[4], Gustavo Reyes-Terán[5], Pablo Tebas[2,3], Luis J. Montaner[2,6], Katharine J. Bar[2,3], Laura A. Vella [2,7,9] ✉ & Michael R. Betts [1,2,9] ✉

Understanding the complexity of the long-lived HIV reservoir during antiretroviral therapy (ART) remains a considerable impediment in research towards a cure for HIV. To address this, we developed a single-cell strategy to precisely define the unperturbed peripheral blood HIV-infected memory CD4+ T cell reservoir from ART-treated people living with HIV (ART-PLWH) via the presence of integrated accessible proviral DNA in concert with epigenetic and cell surface protein profiling. We identified profound reservoir heterogeneity within and between ART-PLWH, characterized by new and known surface markers within total and individual memory CD4+ T cell subsets. We further uncovered new epigenetic profiles and transcription factor motifs enriched in HIV-infected cells that suggest infected cells with accessible provirus, irrespective of reservoir distribution, are poised for reactivation during ART treatment. Together, our findings reveal the extensive inter- and intrapersonal cellular heterogeneity of the HIV reservoir, and establish an initial multiomic atlas to develop targeted reservoir elimination strategies.

HIV reservoir establishment and persistence remains the main barrier preventing a functional cure in PLWH. This reservoir, composed primarily of infected long-lived memory CD4+ T cells[1–3], resides in blood and tissues throughout the body and almost invariably reactivates upon ART interruption[4–7]. Reservoir persistence results from the stable integration of viral DNA into the host cell genome, enabling lifetime infection for the cell and its progeny. Great progress has been made in understanding HIV reservoir virological attributes, including proviral intactness, reactivation potential and integration site landscapes[8,9]; however, beyond superficial characterization and quantification, the cellular identity of the HIV reservoir in vivo remains an enigma. Defining a surface and or epigenetic signature to target the cellular HIV reservoir is of central importance to the HIV cure agenda.

[1]Department of Microbiology, Perelman School of Medicine, University of Pennsylvania, Philadelphia, PA, USA. [2]Center for AIDS Research, University of Pennsylvania, Philadelphia, PA, USA. [3]Department of Medicine, Perelman School of Medicine, University of Pennsylvania, Philadelphia, PA, USA. [4]Centro de Investigación en Enfermedades Infecciosas, Instituto Nacional de Enfermedades Respiratorias, Mexico City, Mexico. [5]Institutos Nacionales de Salud y Hospitales de Alta Especialidad, Secretaría de Salud de México, Mexico City, Mexico. [6]The Wistar Institute, Philadelphia, PA, USA. [7]Division of Infectious Diseases, Children's Hospital of Philadelphia, Philadelphia, PA, USA. [8]Present address: Institute for Medical Engineering and Science, Department of Chemistry, and Koch Institute for Integrative Cancer Research, Massachusetts Institute of Technology, Cambridge, MA, USA. [9]These authors jointly supervised this work: Laura A. Vella, Michael R. Betts. ✉e-mail: vellal@chop.edu; betts@pennmedicine.upenn.edu

**Table 1 | Donor information**

| Stage | Donor | Sex | Age (years) | Tissues profiled | Plasma viral load (log copies ml–1) | Time on ART before trial start (years) |
|---|---|---|---|---|---|---|
| In vitro infection | ND492 | M | 37 | PB | – | – |
| Chronic infection[a] | C01 | M | 33 | Inguinal LN | 6.8 | – |
| | C02 | M | 33 | Cervical LN | 4.88 | – |
| ART-treated infection[a] | A01 | M | [b] | PB | <1.6 | 3.6 |
| | A08 | M | [b] | PB | <1.6 | 4.2 |
| | A09 | M | [b] | PB | <1.6 | 5.6 |
| | B45 | M | 43 | PB | <1.6 | – |

[a]All donors were reported to be infected with HIV group M, subtype B viruses. [b]Age was not reported at the participant level for the original study. However, the age range was reported as between 34 and 52 years.

The main impediment to our understanding of the cellular reservoir is the challenge of identifying infected cells. On average, during ART, fewer than 1 per 1,000 CD4[+] T cells are infected with HIV in the blood[10], most of which have little-to-no viral RNA production[11,12]. To overcome this, studies have used in vitro models[13,14], exogenous stimulation strategies[15,16], clonal enrichment analysis[17] and marker-targeted analysis (many studies[18–20]) to identify surface protein and/or transcriptomic properties of the HIV reservoir. These studies have made inroads into the identity of infected cells but have been limited by the systems employed. These limitations, including (1) infected CD4[+] T cell heterogeneity within bulk or intraclonal populations, (2) incomplete representation of the reservoir after sorting for viral RNA[+] and/or protein-expressing cells and (3) activation-induced malleability of the cell surface proteome and underlying transcriptome, have obfuscated precise characterization of infected cells and highlight the need to develop new strategies to characterize the cellular HIV reservoir at steady state.

Here, we used the presence of integrated provirus as a molecular beacon to directly identify infected cells from treated and untreated PLWH using a single-cell assay for transposase accessible chromatin (scATACseq)[21] and employing a human–viral alignment strategy first developed in a chimeric antigen receptor (CAR) T cell model[22]. scATACseq provides unbiased, single-cell epigenetic profiles and can therefore assign cellular identity to cells with and without detectable provirus. We further combined scATACseq with cell surface protein identity using the recently described ATAC with select antigen profiling by sequencing (ASAPseq)[23,24] to discover cell surface protein signatures that may identify or enrich infected cells. Our findings provide the first direct and unbiased identification of experimentally and in vivo infected memory CD4[+] T cells at single-cell resolution with cellular identity. Our findings also highlight the heterogeneity of HIV[+] cells and the variation of surface and epigenetic markers between and across infection contexts. Together, our extensive characterization of HIV infection using ASAPseq results in a multimodal single-cell dataset of cell surface protein composition, epigenetic landscape and potential regulatory mechanisms to accelerate our understanding of the HIV reservoir for targeted therapy.

## Results

### ASAPseq identification of HIV-infected cells in vitro

To establish the potential for ASAPseq to detect in vitro HIV-infected cells (Table 1), CD4[+] T cells from an HIV-uninfected individual were activated with anti-CD3, anti-CD28/CD49d and interleukin (IL)-2 for 2 days, infected with HIV-1 (molecular clone, SUMA), and rested subsequently for 4 days. We next performed ASAPseq, obtaining 7,095 cells

that passed both ATAC and antibody-derived tag (ADT) component quality checks (Supplementary Table 1 and Extended Data Fig. 1a). Following sequencing alignment to both human and viral genomes, 1,323 cells (18.6%) contained an integrated HIV genome (HIV[+]; Fig. 1a,b). A smaller proportion of the infected cells expressed p24 protein (5.1%; Extended Data Fig. 1b), consistent with known discrepancies between HIV integration and p24 production[25]. To confirm provirus detection specificity, we aligned an unrelated ASAPseq dataset of uninfected peripheral blood (PB) mononuclear cells (PBMCs)[23] using chimeric human–virus genomes (human (hg38) + SUMA HIV and hg38 + HXB2 HIV). No cells contained reads aligning to either HIV genome (Extended Data Fig. 1c). Within the HIV[+] CD4[+] T cells, proviral reads spanned the viral genome but were most prevalent in long terminal repeats (LTRs) as well as in *gag* and *env* (Extended Data Fig. 1d). Fewer reads spanned *pol*, consistent with previous findings[26]. A total of 28% of HIV fragments were in LTR regions, 0.01% spanned both LTR and internal (that is, not LTR) and the remaining fragments (71%) were internal.

We next examined the phenotypic identity of HIV[+] versus HIV[−] cells. We clustered the cells using the epigenetic data and annotated based on imputed gene scores from chromatin accessibility and the ADT component (Fig. 1a, Extended Data Fig. 1e, Extended Data Fig. 2 and Supplementary Table 2). Based on the previous CD4[+] T cell enrichment, most clusters were composed of CD4[+] T cells, with some B and antigen presenting (APC) cells. Most HIV[+] cells (78.6% of HIV[+] cells; Fig. 1b–d) were found within activated and/or effector memory CD4[+] T cell clusters. We observed some HIV[+] cells in other CD4[+] T cell phenotypes (18.3%) and a small proportion in monocytes (1.8%). The presence of virus within monocytes could be genuine given the ability for in vitro infection of monocytic cells[27], or result from the phagocytic and antigen presentation capabilities of B and monocytic cells. These detections could also be multiplets or have insufficient data for correct calling. Regardless of origin, we excluded these cells from downstream analyses and focused on cells bearing definitive T cell surface and epigenetic signatures. These findings reflect the known phenomenon of preferential in vitro HIV infection of activated CD4[+] T cells and demonstrate that this feature can be resolved on a single-cell level by ASAPseq.

Comparing all HIV[+] and uninfected T cells, we observed significantly higher expression of many surface markers, including those associated with T cell activation on HIV[+] cells (Extended Data Fig. 3a and Supplementary Table 3). The top five upregulated markers by π-score[28] on HIV[+] cells included CCR5, SLAM, PD-1, CD49a and CD161, whereas the top five upregulated markers on HIV[−] cells were CD31, CD62L, CD55, CD27 and CD7 (Extended Data Fig. 3a). The significant upregulation of CCR5 in HIV[+] cells is consistent with CCR5 usage as a HIV-1 coreceptor[29]. The other differentially expressed surface markers in HIV[+] cells reflect the known preferential infection of activated CD4[+] T cells[30,31]. We therefore examined whether any surface protein distinguished infected from uninfected cells, specifically in activated clusters (Fig. 1e, Extended Data Fig. 4 and Supplementary Table 3). In activated HIV[+] cells, we found that CCR5, SLAM, CD2, GPR56 and PD-1 were the top upregulated markers, whereas CD62L, CD127, CD49f, CXCR5 and CCR4 were increased in activated HIV[−] CD4[+] cells. This suggests that, even within the activated/effector CD4[+] clusters, HIV[+] cells have higher expression of CCR5 and other activation markers. In the early differentiated HIV[+] T cells, we found significantly increased expression of ICOS, CD25, CXCR3, CD11a and CD49b, whereas the early differentiated HIV[−] cells were enriched in CD62L, CD55, CD127, CD162 and CD45RA (Extended Data Fig. 3b and Supplementary Table 3).

### Epigenetics and ADT predictions of in vitro infected cells

We next examined potential genomic accessibility differences between activated HIV[+] and HIV[−] cells. We observed heightened accessibility in several genomic regions in activated HIV[+] cells (Fig. 2a,b and Supplementary Table 4). We observed greater accessibility at a peak around 1 kb upstream of the *CCR5* transcription start site (Fig. 2c) in activated

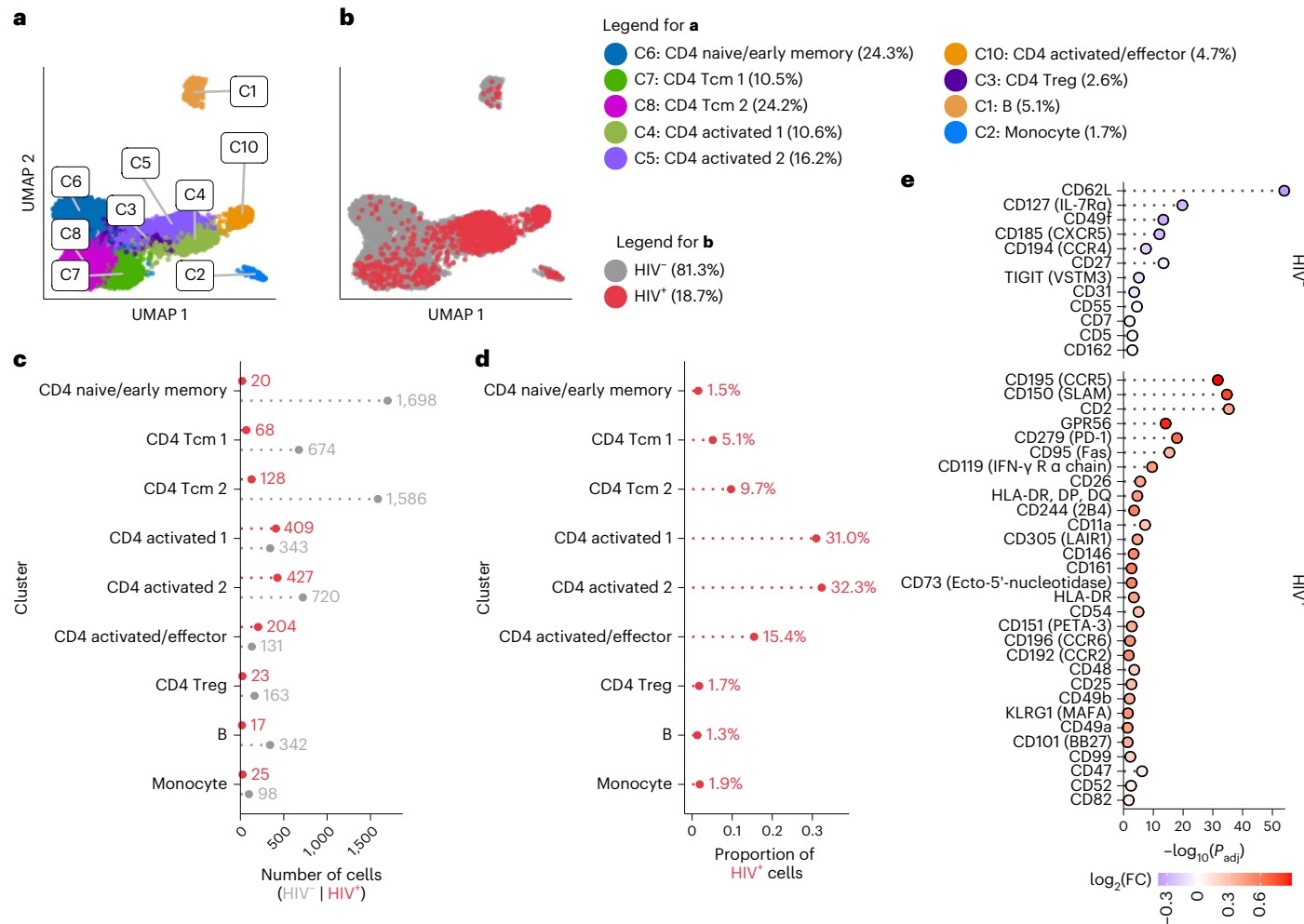

**Fig. 1 | ASAPseq identification of HIV-infected cells in vitro. a**, UMAP representation of ATAC component colored by manually annotated cell phenotypes. **b**, UMAP representation of ATAC component colored by detection of HIV reads. **c**, Absolute count of cell numbers based on annotated clusters. **d**, Percentage of HIV+ cells found in each annotated cluster. **e**, Differential expression of surface antigens was assessed (DESeq2 method in Seurat; two-sided with multiple comparison adjustment using the Bonferroni–Hochberg

method) between activated HIV+ cells ($n = 1,040$) and activated HIV− cells ($n = 1,194$). Positive fold change (FC) values indicate higher expression in HIV+ cells, whereas negative FC values indicate higher expression in HIV− cells. Activated cells are defined as the combination of CD4 activated/effector, CD4 activated 1, and CD4 activated 2 clusters. Markers are arranged from top to bottom by π-score, which is defined as ($-\log_{10}(FDR) \times \log_2(FC)$). All markers shown have an adjusted $P$ value ($P_{adj}$) <0.05.

HIV+ cells, showing a potential concordance between CCR5 surface expression, gene accessibility and infection. From these differentially accessible genomic regions, we assessed transcription factor motif enrichment. In activated HIV− cells (Fig. 2d), the top enriched motifs included several members of the TCF family, including TCF7 (TCF-1) and LEF1, regulators of CD4+ T cell development and differentiation[32]. In activated HIV+ cells (Fig. 2d), top motifs included proliferation and activation motifs in the AP-1 related family such as the JUN and FOS families[33]. These findings, in combination with the surface marker profiles, are concordant with the known immunobiology of HIV-1 infection.

Finally, we employed several supervised machine learning methods to determine whether any cell surface protein combinations could predict infection likelihood. We assessed logistic regression, naive Bayes and random forest (RF) methods using a 70/30 (training/test) split of our dataset. Using all CD4+ T cells, we found similar results between the models with area under the receiver operating curve (ROC) (AUC) values of 0.89, 0.88 and 0.9 respectively (Fig. 2e) suggesting a reasonable classification of HIV+ cells in vitro with this current ADT panel. In the logistic regression, markers with the greatest odds ratio and significance included CD2, CD49b, CD95, SLAM and T cell receptor (TCR) α/β for classifying HIV+ cells while CD4, NKG2D, CD62L and CD162 were

the top markers for classifying HIV− cells (Fig. 2f). For the RF model, the top markers by importance (measured as mean decrease in accuracy when permuted) were CCR5, SLAM, CD49b, CD2 and CD95. When applied to only activated CD4+ T cells, the RF model and logistic regression both performed the best with an AUC of 0.78, compared with 0.74 with naive Bayes. Compared with the total CD4+ T cell models, there were higher false positive rates suggesting that additional markers tailored to activated T cells might benefit classification efforts (Fig. 2g). The activated CD4+ T cell RF was driven primarily by CCR5, SLAM, CD69, CD2 and GPR56 in decreasing order of importance. Overall, these supervised machine learning models are in concordance with our differential expression testing and highlight the prominent roles of CCR5, SLAM and CD2 as markers of HIV-1 in vitro infection (Fig. 2h).

**ASAPseq analysis of infected cells from untreated PLWH**
We next assessed whether HIV-infected memory CD4+ T cells from lymph nodes (LN)—a tissue with a high burden of infected CD4+ T cells[4,5]—of viremic PLWH (Table 1) exhibited bias towards specific cell surface markers or epigenetic characteristics. We enriched memory CD4+ T cells via bead selection from inguinal (C01) or cervical (C02) LN ($n = 2$) and conducted ASAPseq, aligning to a chimeric hg38 + HXB2

genome. In one individual (C01), we profiled 2,554 cells passing quality checks for ATAC and ADT datasets, including 13 HIV+ cells (0.51%). In the second individual (C02), we profiled 6,521 cells passing quality checks, with 41 HIV+ cells (0.63%) leading to a combined chronic dataset of 9,075 cells with 54 HIV+ cells (0.60%; Table 2 and Extended Data Fig. 5a). Most sequenced fragments were internal (74%) followed by LTR only (23%) and spanning both LTR and internal (2%) (Extended Data Fig. 5b). We also noticed a general occlusion of fragments from the *pol* region as observed from the in vitro dataset.

To assess the phenotype of HIV+ cells in LN, we performed clustering into a reduced dimensional form (Extended Data Fig. 5c) and annotated clusters using both ADT and ATAC datasets (Fig. 3a, Extended Data Fig. 6 and Supplementary Table 5). Most profiled cells were memory CD4+ T cells (70%), with the dominant subcluster being CD4+ T-follicular helper (Tfh) cells (19.2%) and a few contaminating clusters including APC, B and follicular dendritic cells (Fig. 3a).

Of the 54 HIV+ cells detected, 64.8% were CD4+ Tfh cells, with the rest identified as CD4+ T resident memory (Trm) and/or T central memory cells (Tcm) (16.7%), APCs (11.1%) and CD4+ T regulatory cells (Treg; 7.4%) (Fig. 3b–d). This distribution corresponds with the known localization of infected cells within germinal centers of secondary lymphoid organs, predominantly in the Tfh compartment[6,34]. While follicular

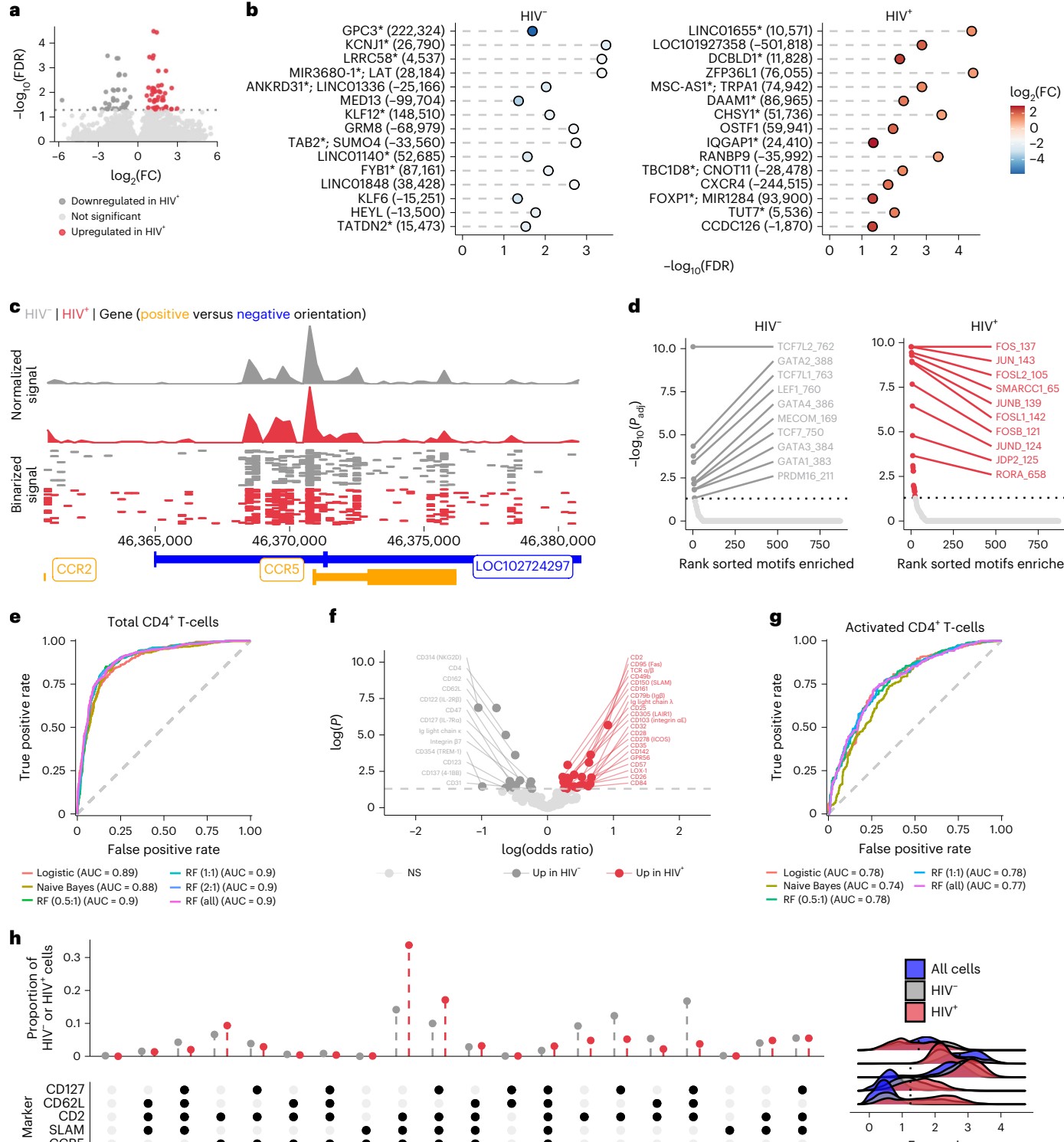

## Table 2 | Cell counts stratified by individual

| Individual | Total cells | HIV⁺ cells (percentage of total cells) | Total HIV DNA copies per million CD4⁺ T cells (percentage of CD4⁺ T cells)[65] |
|---|---|---|---|
| C01 | 2,554 | 13 (0.51%) | – |
| C02 | 6,521 | 41 (0.63%) | – |
| A01 | 14,021 (pre-ATI) 27,065 (post-ATI) | 9 (0.06%; pre-ATI) 6 (0.02%; post-ATI) | 185 (0.019% pre-ATI) 293.8 (0.029% post-ATI) |
| A08 | 18,427 (pre-ATI) 17,461 (post-ATI) | 46 (0.25%; pre-ATI) 36 (0.22%; post-ATI) | 1,791.2 (0.18% pre-ATI) 1,564.5 (0.16% post-ATI) |
| A09 | 44,331 (pre-ATI) 32,998 (post-ATI) | 67 (0.15%; pre-ATI) 36 (0.11%; post-ATI) | 1,297.3 (0.13% pre-ATI) 1,221.8 (0.12% post-ATI) |
| B45 | 12,054 | 10 (0.08%) | – |

dendritic cells were detected, we found no viral DNA in this cluster, even with the likely trapping of infectious virions by follicular dendritic cells[35,36], suggesting high specificity in detecting only HIV DNA.

We next assessed whether HIV⁺ LN CD4 T cells expressed differential surface antigens compared with HIV⁻ LN CD4⁺ cells. CD71, ICOS, HLA-DR, PD-1 and CD151 were preferentially upregulated on HIV⁺ cells, whereas CD48, CD49f and CD47 were upregulated on HIV⁻ cells (Fig. 3e and Supplementary Table 6). As several of these markers are known to be elevated on Tfh cells, we next asked whether any surface marker was enriched for HIV⁺ versus HIV⁻ cells within the Tfh compartment (Fig. 3f). CD71 was the only marker significantly upregulated on HIV⁺ Tfh cells compared with HIV⁻ Tfh cells, suggesting that heightened activation and cycling of individual cells within the Tfh subset is associated with preferential infection[37]. Within the HIV⁻ Tfh cells, we observed higher expression of CD69 and CD45. Together, these analyses suggest that the HIV⁺ cells in LN of untreated PLWH are more activated/cycling, but that the HIV⁺ Tfh cell subset is otherwise quite similar to HIV⁻ Tfh cells in the surface markers assessed.

To define potential differences between the HIV⁺ and HIV⁻ LN CD4⁺ T cells at the epigenetic level, we used chromVAR, specifically designed for differential motif analysis from sparse scATACseq data[38] on all HIV⁺ CD4⁺ T cells versus HIV⁻ CD4⁺ T cells. We found several motifs that had significant bias-corrected deviations between HIV⁺ CD4⁺ cells and HIV⁻ CD4⁺ cells (Fig. 3g). In the HIV⁻ CD4⁺ cells, we found significant motif signatures for the GATA transcription factor family that had increased genomic accessibility (Fig. 3h). In the HIV⁺ CD4⁺

T cells, the most significant motifs showing increased accessibility included POU2F3, FOSB, JUND and BACH1 (Fig. 3i). The presence of AP-1 related transcription factor (including FOS and JUN family) motifs corresponded with the observed increase in activation surface markers, as observed in the in vitro dataset.

Finally, we examined whether particular combinations of surface markers were preferentially associated with infection in viremic PLWH. As before, we split all CD4⁺ T cells into a 70/30 (training/test) split by infection status. Overall, the models performed less well compared with the in vitro analysis. For the chronic samples, the RF model attained an AUC value of 0.79 without downsampling, while the naive Bayes model performed with an AUC value of 0.83 (Fig. 3j). The top markers of importance for the RF model without downsampling include CD4, PD-1, CD71, ICOS, CD200 and CD115, which are generally in concordance with our differential expression testing of surface markers and suggest that the pool of infected cells in LN during chronic infection is highly heterogeneous.

### ASAPseq analysis of infected cells from ART-treated PLWH

We next applied ASAPseq to peripheral blood CD4⁺ T cells in the setting of ART to assess the HIV reservoir in the most accessible compartment and in a clinically relevant context. We selected four fully suppressed ART-PLWH, including longitudinal samples from three participants (A01, A08, A09; pre- and postanalytical treatment interruption (ATI)) from the A5340 ACTG clinical trial who received broadly neutralizing antibody (VRC01) and experienced differential levels of viral rebound during ATI[39] (Table 1). All tested samples were collected while viral load was fully suppressed with ART, as post-ATI samples were taken after ART reinitiation. Given the rarity of infected cells under ART[3] and the paucity of infected cells in blood compared with tissues, we increased the number of memory CD4⁺ T cells assessed, acquiring a combined ART-treated dataset containing 166,357 total cells, of which 213 (0.13%) cells were HIV⁺ (Extended Data Fig. 7a and Table 2). For all donors, we aligned the ATAC reads to autologous HIV sequences (near full length (NFL) or HIV env) and HXB2 to increase the odds of detecting infected cells due to the prevalence of proviral mutations in long-term PLWH[8]. With the autologous alignments, most reads were detected in the LTR and/or nef region (Extended Data Fig. 7b). Cells from donors A08 and A09 also had internal fragments that mapped to gag, pol or env genes (Extended Data Fig. 7b).

After clustering based on the ATAC component and annotating with both ADT and ATAC components (Extended Data Fig. 8 and Supplementary Table 7), we identified multiple memory CD4⁺ T cell subsets with only small numbers of contaminating cells (Extended Data Fig. 7c and Fig. 4a,b). The T cell clusters separated based on epigenetic profiles associated with T cell differentiation states. For example, Tcm had distinct epigenetic and surface antigen profiles compared with those seen in effector/effector memory (Tem) cells (Extended Data Fig. 8). Across the ART-PWLH, we found no consistent predominately infected cell type.

**Fig. 2 | Differential chromatin accessibility in HIV-infected cells and surface marker based supervised machine learning in vitro. a**, Volcano plot showing differentially accessible peaks between activated HIV⁺ (n = 1,040) versus activated HIV⁻ (n = 1,194) cells. **b**, Top 15 significant peaks in activated HIV⁻ versus activated HIV⁺ cells from **a**. Peaks are ranked by π-score (−log₁₀(FDR) × log₂(FC)); y axis labels denote nearest TSS and/or whether the peak is in a gene (marked by an asterisk) and numbers in parentheses indicate distance to the nearest TSS. Negative numbers indicate that the nearest TSS is upstream of the peak. **c**, Top, Aggregated and normalized ATAC signal between activated HIV⁻ and activated HIV⁺ cells at the CCR5 locus. Middle, Binarized ATAC signal at a single-cell resolution (showing random 750 cells for each group). Bottom, Gene map for the genomic region shown: chr3:46360853–46380854 (centered on CCR5). **d**, Significant motif enrichments found in the differential peaks (ordered by significance) are shown in activated HIV⁻ cells or activated HIV⁺ cells (two-sided Wilcoxon rank sum test with multiple comparison correction using Benjamini–

Hochberg method in getMarkerFeatures function of ArchR). **e**, AUC plots for multiple supervised machine learning models for all CD4⁺ T cells. Ratios for different RF models indicate the number of HIV⁻ cells used for each HIV⁺ cell (that is, 5:1 ratio meant that the HIV- cells in the training dataset were randomly downsampled to get only five times the number of HIV⁺ cells in the training dataset). **f**, Significant (two-sided z test) coefficients for the logistic regression model shown in **e**. Positive odds ratio indicates a marker has more weight for HIV⁺ cells while negative odds ratio indicates a marker has more weight for HIV⁻ cells. NS, nonsignificant. **g**, AUC plots for multiple supervised machine learning models for activated CD4⁺ T cells. **h**, Proportion of HIV⁻ and HIV⁺ activated CD4⁺ T cells with different surface marker combinations. Bottom dot plot indicates the combination of positive markers, which was determined by the thresholds indicated with the dotted line on the ridge plots. These ridge plots display the expression distribution of activated HIV⁺ CD4⁺ T cells, activated HIV⁻ CD4⁺ T cells and total cells in the in vitro dataset to help with gating.

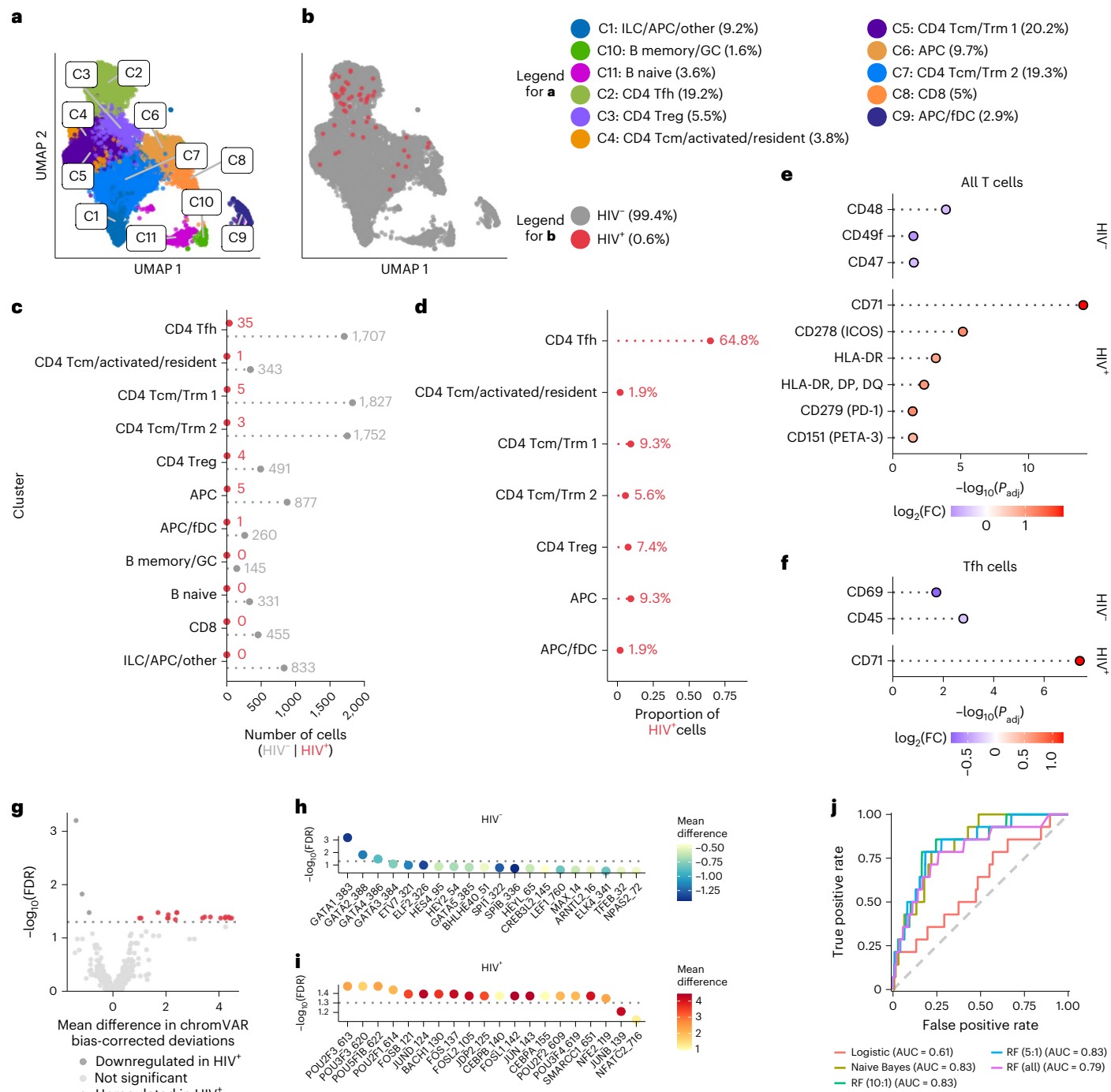

**Fig. 3 | ASAPseq identification of HIV-infected cells predominantly in Tfh cells from lymph nodes of untreated PLWH. a**, UMAP representation of ATAC component colored by manually annotated cell phenotypes. **b**, UMAP representation of ATAC component colored by detection of HIV reads. fDC, follicular dendritic cells. **c**, Absolute count of cell numbers based on annotated clusters. **d**, Percentage of HIV+ cells found in each annotated cluster. **e**, Differential expression of surface antigens was assessed (DESeq2 method in Seurat; two-sided with multiple comparison adjustment using Bonferroni–Hochberg method) between HIV+ CD4+ T cells versus HIV− CD4+ T cells. **f**, Same as in **e** but for HIV+ Tfh cells versus HIV− Tfh cells. Markers are ranked in **e**–**f** by

π-score (see Fig. 1e legend). All markers shown have a $P_{adj}$ value < 0.05. **g**, Comparison of motifs associated with accessible chromatin regions of HIV+ CD4+ T cells versus HIV− CD4+ T cells. The volcano plot displays the differentially enriched motifs from chromVAR and ArchR, with a threshold of FDR <0.05 indicating statistical significance. Mean difference values greater than zero indicate an enrichment in HIV+ cells, while negative mean difference values indicate enrichment in HIV− cells. **h,i**, The top 20 motifs (ordered by FDR) are shown for CD4+ HIV− T cells (**h**) and CD4+ HIV+ T cells (**i**). **j**, AUC plots for various supervised machine learning methods. Numbering format for RF models is explained in Fig. 2.

Some donors (A01 and B45) showed focusing of HIV+ cells within a single cell type, whereas other donors (A08 and A09) were more diverse (Fig. 4c). After aggregating all HIV+ cells across donors, we detected HIV+ cells within multiple subsets including CD4+ Tcm, Th2, circulating Tfh (cTfh), mucosal associated invariant T (MAIT) cells and Tem/effector cells (Fig. 4c,d). We also detected a small proportion of HIV+ cells within the contaminating APC clusters, but it is unclear whether these were in actual APCs, CD4+ T cells included in doublets with APCs or misidentified as APCs.

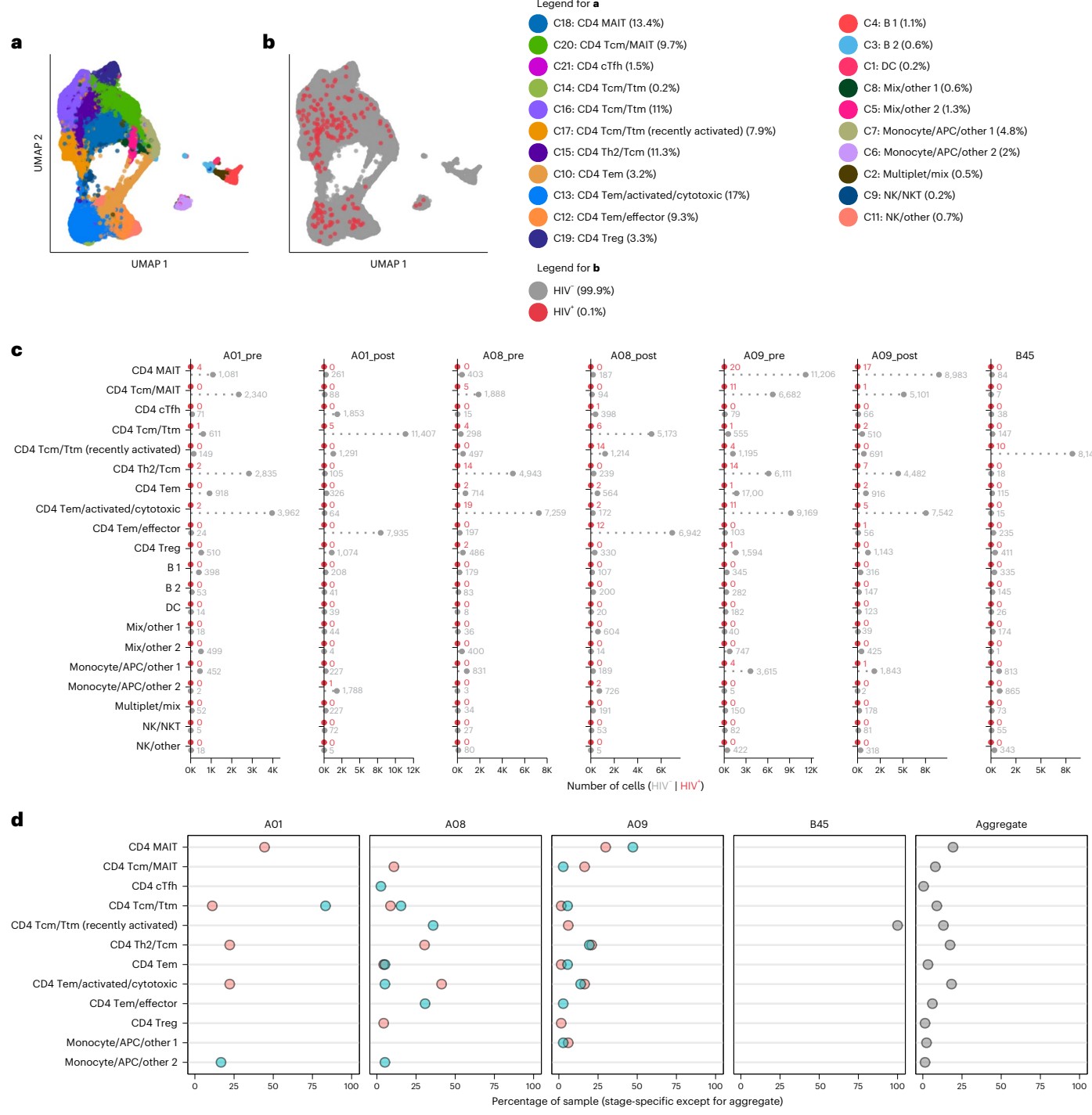

**Fig. 4 | ASAPseq identification of heterogeneous HIV-infected cells from peripheral blood of ART-suppressed PLWH. a**, UMAP representation of ATAC component colored by manually annotated cell phenotypes. **b**, UMAP representation of ATAC component colored by detection of HIV reads. **c**, Absolute count of cell numbers based on annotated clusters. **d**, Percentage of HIV+ cells found by cluster separated by donor and whether the sample was collected before or after ATI or not applicable (B45 only); *x* axis represents the percent of HIV+ cells in each specific sample that were found in each annotated cluster. The right panel indicates the aggregate values across the entire ART-treated dataset.

We next assessed reservoir compositional stability after ATI in longitudinal samples from A01, A08 and A09. We observed maintenance of infected cell subsets between the pre- and post-ATI timepoints from donor A09 (Fig. 4c,d) who had a low viral rebound during ATI[39], suggesting relative stability of the HIV reservoir cellular phenotypes after ATI. In contrast, individual A01 had a moderate degree of phenotype maintenance, while A08 demonstrated more prominent reservoir modulation with the appearance

of infection within recently activated Tcm/T transitional memory (Ttm) cells at the post-ATI timepoint. Both A08 and A01 had higher levels of viral rebound compared with the level of viral rebound in A09 (ref. [39]).

### Heterogeneity of HIV-infected cells in blood of ART-PLWH
We next assessed differential surface marker expression between circulating HIV+ CD4+ T cells and HIV− CD4+ T cells in ART-PLWH using several

statistical methodologies. Using Seurat and DESeq2, we found significantly higher expression of CCR5, PD-1 and CD2 in HIV+ cells, markers that are associated with activation and costimulation. We found higher expression of several markers in the HIV− T cells including CD74, CD41, NKG2D and KIR3DL1 (Fig. 5a and Supplementary Table 8). By Wilcoxon analysis, we found additional activation markers more highly expressed on HIV+ CD4+ T cells including SLAM, CCR5, PD-1, CD49d, HLA-DR and CCR2 (Fig. 5b, Extended Data Fig. 9 and Supplementary Table 8). Similar to the results from the DESeq2 method, we found that CD305 were expressed more on HIV− CD4+ T cells. Notably, the relative increase in expression of some of these markers (from either test) was small and varied across donors as seen with PD-1 in HIV+ cells being driven primarily by individual A08 (Extended Data Fig. 9).

To reduce the variance caused by differential expression of markers based on memory and functional differentiation (that is, Tcm/Ttm, Tem/effector and MAIT cells), we conducted differential expression testing between HIV+ and HIV− cells in the following aggregated CD4+ T cell groups: Tcm/Ttm cells (Fig. 5c), Tem/effector cells (Fig. 5d) and MAIT cells. Other cell types were excluded from this analysis due to low HIV+ cell counts. In the Tcm/Ttm cells, we found increased expression of CD2, CCR5, CD11a, CD26, CD71, CD99 and CD18 on HIV+ cells, whereas there was increased expression of CD74, CD79b, CX3CR1, LOX-1 and CD62L on the HIV− cells, indicating that the Tcm/Ttm cells contributed to the differential CD2 expression within the bulk CD4+ T cell analyses (Fig. 5c). In the Tem/effector cells, PD-1, HLA-DR and CCR5 were expressed more on HIV+ cells, while multiple markers including CD41, CD305 and CD101 were more expressed on HIV− Tem/effector cells (Fig. 5d). These results highlight the surface marker and phenotypic heterogeneity present in specific memory phenotypes as well as the inability of any marker amongst the 154 antibodies tested to uniquely identify HIV+ cells.

To address this heterogeneity in a more unbiased way, we assessed whether there were unique epigenetic signatures between the HIV+ and HIV− CD4+ T cells using chromVAR[38] on all HIV+ versus HIV− cells grouped by the same three aggregate memory/functional phenotypes (Tcm/Ttm, Tem/effector and MAIT). We identified several highly enriched motifs in the different groups over all other cells that were not in the analyzed group (collectively referred to as background cells). Motifs were clustered into different modules via hierarchical clustering (Fig. 5e). In support of our manual annotation, Tcm/Ttm cells showed greater enrichment of Tcf7 and Lef1 motifs (module 1) compared with Tem/effector cells, while Tem/effector cells showed enrichment of Tbx21 (T-bet), Eomes and interferon related motifs (modules 4, 7 and 8). MAIT cells, which can be heterogeneous in terms of different memory phenotypes, showed enrichment in motifs from

the nuclear factor-kappaB (NF-κB) transcription factor family including NFKB1, REL, RELA and RELB (module 3).

While HIV+ and HIV− cells shared similar motifs within the aggregate clusters, we observed patterns of greater enrichment of certain motifs in the HIV+ cells. Modules 9 and 10—which include BACH2 and AP-1 related family transcription factors such as Jun and Fos—were highly enriched for HIV+ Tcm/Ttm and Tem/effector cells compared with all other cells as background. When specifically comparing Tcm/Ttm HIV+ versus Tcm/Ttm HIV− cells, we found significant enrichment of these motifs (Fig. 5f). HIV+ MAIT cells curiously did not show enrichment in this module but did show a trend towards increased accessibility in module 3 motifs (Fig. 5e). Altogether, these motif signatures across our dataset indicate that Tcm/Ttm and Tem/effector HIV+ cells share an epigenetic signature consistent with a constitutively heightened immune activated state.

## Discussion

Targeted elimination of the HIV reservoir requires a deep understanding of both the virologic and cellular characteristics of infected cells. While previous work has yielded extensive knowledge of the virologic properties of the viral reservoir, precise characterization of the cellular reservoir has proven more difficult due to infected cell rarity and lack of readily measurable defining characteristics. Here, we applied a new single-cell genomic and bioinformatic ASAPseq strategy to identify infected cells using integrated proviral DNA in the multiomic context of single-cell epigenetic and surface antigen profiling. We found that cells with proviral DNA can indeed be detected and individually characterized directly ex vivo even from the most challenging scenario of peripheral blood of PLWH on ART. Our strategy has broad application for in depth analysis of the HIV reservoir within different cell populations, therapeutic interventions and HIV cure and eradication studies[40].

Numerous studies have attempted to identify cell surface markers that enrich or specifically identify CD4+ T cells harboring integrated proviral DNA. We profiled many individual markers previously associated with HIV infection, including PD-1 (ref. [34]), CXCR5 (ref. [34]), CCR6 (ref. [41]), CTLA-4 (ref. [42]), CD69 (ref. [43]), OX40 (ref. [44]), CD2 (ref. [45]), Lag-3 (ref. [19]), TIGIT[18,19], CD20 (ref. [46]) and CD161 (ref. [47]). No single previously identified cell surface marker was associated solely with, or predictive of, the presence of integrated provirus. However, specific combinations of markers, either by presence or relative level, are more reliably enriched for HIV-infected cells depending on the cells profiled. For example, CCR5 was a top differential marker between HIV+ and HIV− cells for in vitro infected cells and ART-PLWH, but was not heightened on HIV+ cells in LN CD4+ T cells of untreated PLWH. Instead, activation and Tfh associated markers (CD71, ICOS, HLA-DR

**Fig. 5 | High degree of heterogeneity in HIV-infected cells from peripheral blood of ART-suppressed PLWH. a**, Differential expression of surface antigens was assessed between all HIV+ CD4+ T cells (n = 205) versus HIV− CD4+ T cells (n = 146,016). Test performed using the DESeq2 pseudobulk method in Seurat (two-tailed with multiple comparison adjustment using the Bonferroni–Hochberg method). **b**, Same comparison and cells as in **a** but using the Wilcoxon statistical test in Seurat (two-sided with multiple comparison adjustment using the Bonferroni method). **c**, Differential expression of surface antigens was assessed between Tcm/Ttm HIV+ (n = 57) versus Tcm/Ttm HIV− cells (n = 39,954) using the DESeq2 method in Seurat (two-tailed with multiple comparison adjustment using the Bonferroni–Hochberg method). Tcm/Ttm cells are defined by the combination of all clusters containing the terms 'Tcm' or 'Ttm' or 'cTfh' (but not MAIT or recently activated Tcm/Ttm cells) in Fig. 4a. **d**, Differential expression of surface antigens was assessed between Tem/effector HIV+ (n = 59) versus Tem/effector HIV− (n = 48,928) cells using the DESeq2 method in Seurat (two-tailed with multiple comparison adjustment using the Bonferroni–Hochberg method). Tem cells are defined by the combination of all clusters containing the term 'Tem' in Fig. 4a. Markers in **a**–**d** are ranked by π-score (see Fig. 1e legend). All markers shown have an adjusted P value < 0.05. **e**, Comparison of motifs associated with accessible chromatin regions of cells grouped by

phenotype (MAIT (n = 58 for HIV+; n = 38,405 for HIV−), Tcm/Ttm (n = 85 for HIV+; n = 53,135 for HIV−; includes recently activated Tcm/Ttm cells) and Tem/effector (n = 59 for HIV+; n = 48,928 for HIV−)) and infection status (HIV− versus HIV+) as assessed through chromVAR and ArchR. Heatmap displays the mean chromVAR deviations (z score) of transcription factor motifs from CISBP database (displayed by row) that define each aggregate group (displayed by column). Motifs were selected by FDR < 0.05 and a mean difference > 0.5, indicating a significant accessibility of regions that contain a given transcription factor motif for cells in the cluster (indicated by asterisks) as compared with all other cells (getMarkerFeatures function in ArchR). Each cell is colored by the mean deviation (z score) where values greater than zero indicate positive enrichment of a motif. Motifs were clustered using k-means clustering and the motifs are labeled to the right of the heatmap in order from top to bottom for each cluster. Asterisks indicate that the motif was significantly enriched in the specific group (column). *P < 0.05; **P < 0.01. **f**, Differential chromVAR motif profiles were assessed between HIV+ Tcm/Ttm cells and HIV− Tcm/Ttm cells. The top 20 most significant motifs for HIV+ cells are shown. The dotted line indicates a −log$_{10}$ transformed FDR value of 0.05. Color indicates the mean difference in chromVAR deviations (value greater than zero indicates enrichment in HIV+ cells).

and PD-1) predominated among the differentially expressed surface proteins within the total LN CD4 population. CD71, correlated with cell cycling[37] and ferroptosis[48], was highly enriched on HIV[+] Tfh cells of untreated PLWH, suggesting that HIV[+] Tfh cells are dividing frequently and potentially predisposed to cell death. CD71 was also enriched on blood HIV[+] CD4[+] Tcm/Ttm cells from ART-PLWH, possibly reflecting a recently proliferating reservoir subset. We also observed CD2 and PD-1 enrichment on HIV[+] T cells from ART-PLWH in Tcm/Ttm HIV[+] T cells and Tem HIV[+] T cells, respectively. Our results are in concordance with

previous studies that showed various markers can enrich (but not homogeneously select) for cells with HIV DNA including CD2 (ref. [45]), PD-1 (ref. [34]), HLA-DR[49] and CD11a[50].

Identification of cell surface markers found preferentially on HIV[−] cells is also useful. We found greater expression of CD48, an NK-cell coactivating ligand for CD244/2B4 (ref. [51,52]), on HIV[−] cells during untreated infection. CD48 can be downregulated by HIV[+] cells during in vitro infection to escape autologous NK-cell-mediated killing[53]. Another marker of interest increased on HIV[−] cells was CD305 (LAIR-1),

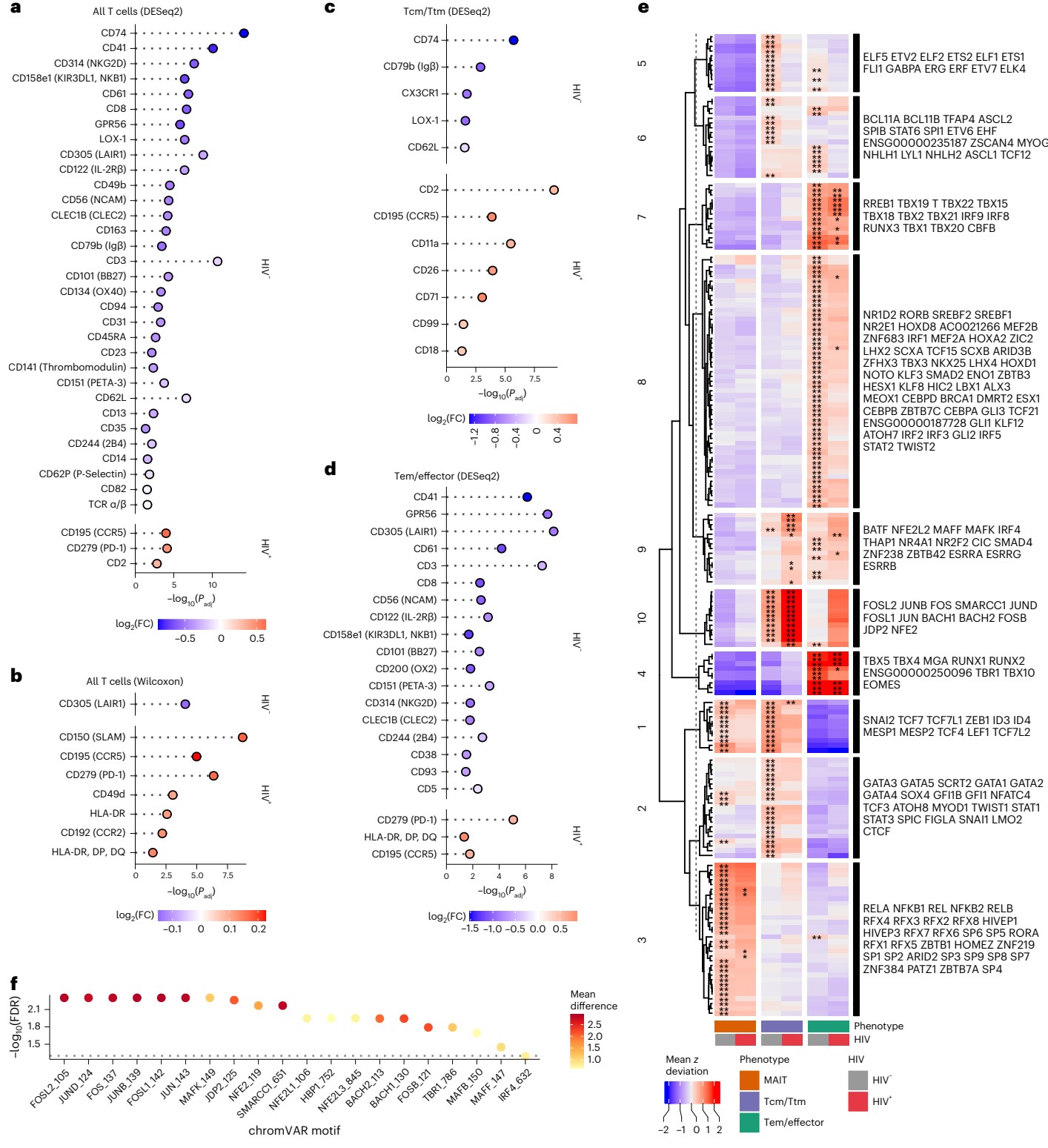

which inhibits activation via the TCR[54]. We also found an increase in CD74 (major histocompatibility complex class II invariant chain) in HIV⁻ Tcm/Ttm cells. Our findings, as a whole, indicate that HIV⁺ cells are highly heterogeneous at interpersonal and intrapersonal levels, and that each cell subset, compartment and infection state needs to be evaluated individually.

Our ASAPseq epigenetic modality yielded insights into both known and potential differential regulatory factors that impact HIV infection in vitro and ex vivo. One resounding theme across in vitro infection, untreated and treated in vivo infection was heightened accessibility for AP-1 family (Fos, Jun) and Bach1/Bach2 transcription factor motifs. The presence of AP-1 motifs suggests that these cells are poised for, or have a history of, activation/proliferation, which may indicate an increased likelihood for viral RNA transcription and reactivation upon ART interruption or treatment with latency reversal agents. Activated in vitro infected HIV⁺ CD4⁺ T cells, which had elevated CCR5 protein expression, also had increased accessibility upstream of the *CCR5* transcription start site. We further identified differentially accessible peaks near genes previously implicated in HIV infection, including *DCBLD1* (increased HIV replication), *GGA2* (Nef-mediated CD4 downregulation) and *PLEKHA3* (increased HIV replication)[55–57]. During chronic infection, we observed increased transcription factor motif accessibility, including POU2F1 (Oct-1) and POU2F2 (Oct-2), previously shown to repress the HIV-1 LTR promoter[58]. These indicate a possible role in suppressing HIV activity and maintaining longevity of infected cells. Furthermore, the GATA1 motif was one of the most significant accessible motifs in HIV⁻ cells, which is striking given its role in repressing CCR5 expression in human CD4⁺ T cells[59]. Taken together, our epigenetic analysis provides a framework and new opportunities for understanding the complex regulation of HIV infection and ongoing processes regulating uncontrolled infection.

Recent reports have suggested that Tem/effector cells are the prominent memory pool with active proviral transcription during ART[11,60,61]. However, we find no Tem bias in the cells harboring integrated virus in ART-PLWH. The lack of memory bias may reflect the true heterogeneity of HIV-infected cells that constitute the HIV reservoir, or might be attributed to differences in when ART was initiated during infection and the duration of ART[62–64]. We also found that the degree of phenotype stability within the reservoir after ATI during VRC01-bNab therapy may be tied to rebound viral load[39]. Individual A08, who experienced high viral rebound after ATI, demonstrated evidence of reservoir diversification post-ATI with a distinct appearance of virus within both recently activated Tcm/Ttm and Tem populations. Previous studies demonstrated proviral diversification[39,65] in the same individual after ATI. Individual A09, who had the highest level of phenotype stability, had the lowest rebound viral load of the three individuals assessed and displayed VRC01-resistant provirus[39,65]. Until now, this degree of heterogeneity across individuals and longitudinal conservation of phenotypes across ATI has not been demonstrated and indicates the necessity of single-cell based approaches not solely dependent on viral transcription.

The main limitation of ASAPseq to identify infected cells is the requirement for the provirus to be in accessible chromatin, which may bias our surface and epigenetic profiling towards an activated phenotype or the 'active' reservoir[11]. Proviruses become less accessible over time in vitro[66] and after extended ART[67]. Another limitation is the difficulty in assessing proviral intactness due to short-read sequencing and our transposase-based methodology. However, alignments towards multiple proviral genes in a single cell can be used as a proxy for intactness given that most defective proviruses from ART-PLWH have large deletions[8]. We also observed a dropoff in sequence recovery from *pol* regions, possibly explained by nucleosomal occupancy[68] or more persistent transcription factor binding[69]. While it is likely that many of our detections from ex vivo samples are defective proviruses, these cells are still relevant as viral proteins can still be produced[70,71], which

may interface with the immune system and potentially complicate latency reversal studies. As such, our focus on HIV DNA⁺ cells, regardless of intactness or transcription, is still relevant given that these cells survived clearance by the immune system and/or the short half-life associated with HIV infection kinetics[72].

Overall, we have highlighted the complex heterogeneity of the HIV reservoir using a new unbiased genomic strategy to identify HIV infected cells at single-cell resolution with simultaneous surface antigen and epigenetic data. With this unparalleled resolution via ASAPseq compared with currently published strategies for studying the HIV reservoir, we uncovered both known and new surface markers of HIV infection as well as accessible transcription factor motifs that may regulate, or be indicative of, HIV infection. Together, our strategy and data initiate a multiomic atlas of HIV-infected cells with phenotypic and epigenetic characteristics, thus contributing towards the united goal of identifying the HIV reservoir for a targeted HIV cure strategy.

## Online content

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

# Methods

## Study approval

This study was approved by the Institutional Review Boards at the University of Pennsylvania and the University of Alabama at Birmingham. This study complies with all relevant ethical regulations. PLWH ($n = 2$) were originally recruited by the Centro de Investigación en Enfermedades Infecciosas at the Instituto Nacional de Enfermedades Respiratorias (CIENI-INER) in Mexico City, Mexico. All donors provided informed consent for LN tissue donation in compliance with protocols set forth by the Ethics Committee and the Ethics in Research Committee of the INER (study number: B03-16) and the Institutional Review Board at the University of Pennsylvania. ART-treated samples ($n = 3$), A01, A08 and A09, were provided from the ACTG clinical trial A5340, which was conducted with protocols set forth by the Institutional Review Boards at the University of Pennsylvania and the University of Alabama at Birmingham and was previously published[39]. The original clinical trial included provisions for research related to this study. All donors for this study provided informed consent in compliance with protocols set forth by the respective institutional review boards. Another ART-treated PLWH ($n = 1$), B45, was recruited from the BEAT-HIV program cohort where an apheresis sample was collected under ART. All participants were compensated for their time and study visits. No additional compensation was provided for this study.

## Samples

For the in vitro infection model, PBMCs were obtained from an HIV-negative donor apheresis from the Human Immunology Core at the University of Pennsylvania. PBMC were cryopreserved at the time of receipt. For the chronic infection model, inguinal or cervical lymph nodes were obtained from HIV-1[+] individuals before the introduction of ART, and cryopreserved. For ART-treated studies, PB samples were acquired via apheresis and cryopreserved. Samples A01, A08 and A09 had previously experienced therapy interruption through the ACTG A5340 trial, but were under at least 6 months (for the post-ATI timepoint) of ART suppression at the time of analysis. PBMCs were prepared using standard density gradient centrifugation and cryopreserved at −150 °C.

## In vitro infection model

Uninfected PBMC were thawed and rested overnight in a humidified incubator at 37 °C. CD4[+] T cells were negatively enriched (STEMCELL Technologies, catalog no. 19052), pelleted and resuspended in complete RPMI medium at a concentration of $2 \times 10^6$ cells ml$^{-1}$ in a six-well tissue culture plate. An activation cocktail consisting of anti-CD3 (1 μg ml$^{-1}$; Bio-Rad, catalog no. MCA463EL), recombinant IL-2 (100 U ml$^{-1}$; Sigma Millipore catalog no. 11011456001), and anti-CD28 + anti-CD49d (1.5 μg ml$^{-1}$; BD, catalog no. 347690) were added to the cells. The cells were activated for 2 days in a humidified incubator at 37 °C. Cells were then pelleted in a 15 ml tube and resuspended in 1 ml of complete RPMI. HIV-1 virus stock (strain SUMA; provided by the University of Pennsylvania CFAR Virus and Reservoirs Core) was thawed briefly in a 37 °C water bath; 50 ng (p24) of viral stock was added to the suspension and mixed by pipetting. Cells were then infected by spinoculation for 45 min at 400$g$. Cells were rested in a humidified incubator at 37 °C for 1 h (cap loosened to promote gas exchange) before washing with complete RPMI and pelleting. Cells were resuspended at a concentration of $2 \times 10^6$ cells ml$^{-1}$ and left to rest for 2 days in a six-well tissue culture plate. Complete RPMI was added to the cells after 2 days. The cells were collected after an additional 2 days and pelleted for dead cell removal based on Annexin V using the Dead Cell Removal (Annexin V) Kit (STEMCELL Technologies, catalog no. 17899).

## Flow cytometric verification of HIV-1 infection

Staining and flow cytometry were based on a previously published protocol[73]. Approximately 1.5 million cells from the in vitro infection culture were spun down at 400$g$ for 5 min and resuspended in 45 μl PBS. Live/dead staining was performed using 5 μl of a 1:60 dilution stock of prepared Live/Dead Fixable Aqua Dead Cell Stain (Invitrogen). Cells were stained for 5 min in the dark at room temperature. A staining cocktail with fluorescence-activated cell sorting (FACS) buffer and CD8 BV570 (BioLegend, catalog no. 301038, clone RPA-T8; 0.3 μl per test) was added for a 10 min stain in the dark at room temperature. Then, 1 ml FACS buffer was added and the cells were spun down at 400$g$ for 5 min. Cells were permeabilized with 250 μl of BD Cytofix/Cytoperm solution (BD, catalog no. 554714) for 18 min in the dark at room temperature; 1 ml BD Perm/Wash Buffer (BD, catalog no. 554714) was added and cells were spun down at 600$g$ for 5 min. After discarding the supernatant, cells were resuspended in staining solution containing anti-p24 FITC (Beckman Coulter, catalog no. 6604665; clone KC57; 0.1 μl per test) and BD Perm/Wash Buffer for a final staining volume of 50 μl. Cells were stained in the dark for 1 h at room temperature. Cells were washed with 1 ml BD Perm/Wash Buffer and fixed with 350 μl 1% paraformaldehyde. Data were acquired on a BD FACS Symphony A5 cytometer and analyzed using FlowJo (v.10, BD).

## Memory CD4[+] T cell enrichment

Memory CD4[+] T cells were enriched by negative selection using the EasySep Human Memory CD4[+] T Cell Enrichment Kit (STEMCELL Technologies, catalog no. 19157) following the recommended protocol. After collecting enriched memory CD4[+] T cells, the cells were spun down at 400$g$ for 5 min and resuspended in 500 μl PBS. Cells were counted with Trypan blue staining using a Countess II (Invitrogen) before beginning the ASAPseq protocol.

## ASAPseq—cell preparation and staining

Buffer and cell preparation were performed as previously published[23] and as described below. Cells ($5 \times 10^5$ to $1 \times 10^6$) were resuspended in 22.5 μl Staining Buffer and incubated with 2.5 μl TruStain FcX (BioLegend, catalog no. 422302) for 10 min on ice. One test of the TotalSeq-A Human Universal Cocktail, v.1.0 (BioLegend, catalog no. 399907) was prepared according to the manufacturer's protocol with ASAPseq Staining Buffer. We chose this particular cocktail of antibodies because this was the largest commercially available panel and would therefore enable a more unbiased approach to assessing the surface antigen profiles. Each antibody in this pool was pretitrated by BioLegend for use in sequencing-based methods. The antibody cocktail (25 μl) was added to the cells for 30 min on ice. Cells were washed with 1 ml Staining Buffer and pelleted (all spins at 400$g$ for 5 min with centrifuge set at 10 °C) for a total of three washes. Cells were then resuspended in 450 μl PBS; 30 μl 16% paraformaldehyde was added to fix the cells for 10 min at RT with occasional swirling and inversion. The fixation reaction was quenched with 25.26 μl 2.5 M glycine solution. Cells were then washed and pelleted with 1 ml ice-cold PBS for a total of two washes. Fixed cells were permeabilized using 100 μl OMNI lysis buffer for 3 min on ice. Cells were then washed and pelleted with 1 ml Wash Buffer (spin at 500$g$ for 5 min with centrifuge set at 10 °C). After removing the supernatant, cells were resuspended in at least 150 μl (depending on original cell input) of 1× Nuclei Buffer (10x Genomics) and strained to remove aggregates using a 40 μm FlowMi strainer (Sigma, catalog no. BAH136800040-50EA). Cells were counted using Trypan blue staining to verify successful permeabilization (>95% Trypan blue positive staining). Cells were diluted as needed according to the scATACseq Chip H protocol (10x Genomics).

## ASAPseq—library preparation

Single-cell droplets were generated using the Chromium platform and a scATACseq Chip H kit (10x Genomics). Modifications mentioned in the original ASAPseq protocol were performed to allow for capture of ADT[23]. Briefly, during the barcoding reaction (step 2.1), 0.5 μl of 1 μM bridge oligo A (BOA) was added to the barcoding mix. The sequence of the bridge oligo is: TCGTCGGCAGCGTCAGATGTGTATAAGAGACA-GNNNNNNNNNVTTTTTTTTTTTTTTTTTTTTTTTTTTTTTT/3InvdT/. To facilitate BOA annealing during GEM (Gel bead-in Emulsion, 10x Genomics) incubation (step 2.5), a 5 min incubation at 40 °C was added

at the beginning of the GEM amplification protocol. During silane bead elution (step 3.1o), beads were eluted in 43.5 µl of Elution Solution I and 3 µl was kept aside to use as input in the tag library PCR, while the remaining 40 µl was used to proceed with SPRI cleanup as described in the 10x Genomics scATACseq protocol. During SPRI cleanup (step 3.2d), the supernatant was saved and 32 µl SPRI beads was added for TotalSeq-A product isolation. Beads were washed twice with 80% ethanol and eluted in 42 µl Qiagen EB. This fraction was combined with the 3 µl saved from step 3.1o after the silane purification to be used as input in the protein tag indexing reaction. PCR reactions were set up to generate the protein tag library using P5 (AATGATACGGCGACCACCGAGA) and RPI-x primers (see below). Amplification settings were as followed: 95 °C 3 min, 14 cycles of 95 °C 20 sec, 60 °C 30 sec and 72 °C 20 sec, followed by 72 °C for 5 min and ending with hold at 4 °C. The final libraries were quantified using a Qubit dsDNA HS Assay kit (Invitrogen) and a High Sensitivity D1000 DNA tape on a Tapestation D4200 (Agilent).

### RPI-x primers

Indexing primers follow the format: CAAGCAGAAGACGGCATACGAGA-TxxxxxxGTGACTGGAGTTCCTTGGCACCCGAGAATTCCA where xxxxxx refers to the Illumina TruSeq Small RNA indices. Indices 1 through 8 were used in this study.

### ASAPseq−library sequencing

Sequencing runs were performed on NextSeq 550 or NovaSeq 6000 platforms (Illumina) with a target of at least 10,000 reads per cell for ADT libraries and 25,000 reads per cell for ATAC libraries.

### Initial ATAC data processing

Libraries sequenced via the NovaSeq were filtered for potential index hopping by using index-hopping-filter (10x Genomics; v.1.1) in ATAC mode. Filtered reads were then processed the same as other libraries sequenced with the NextSeq. A chimeric genome was built using cellranger-atac mkref (10x Genomics; v.2.0.0) with the default hg38 genome (10x Genomics; refdata-cellranger-arc-GRCh38-2020-A-2.0.0) along with either SUMA_TF1 (for in vitro), HXB2 (for chronic) or autologous sequences plus HXB2 (for ART) (Supplementary File 1 and Supplementary Table 9). The motifs.pfm file from the default hg38 genome reference was copied over to the new chimeric genome to assist with quality control during downstream processing by cellranger. The reads were aligned and counted to their respective chimeric genome using cellranger-atac count. After alignment, barcodes pertaining to multiplets were selected using AMULET[74] (using default parameters were false discovery rate (FDR)-corrected (Bejamini−Hochberg method) P value > 0.01 is denoted as a multiplet) to be filtered out from the dataset. Fragment files from cellranger-atac were loaded into ArchR (v.1.0.2)[75] for downstream analysis. Arrow files were created for each sample. For in vitro and ART-treated samples, cells were filtered for TSS Enrichment greater than or equal to eight while chronic samples were filtered for TSS Enrichment greater than or equal to six. Barcodes that were selected by AMULET as being multiplets were then filtered out of the Arrow file. Samples, by level, continued in the ArchR pipeline. Briefly, iterative latent semantic indexing, Harmony (for batch effect correction as needed), cluster generation (via Seurat) and uniform manifold approximation and projection (UMAP) generation were performed. All specific parameters and settings are documented in our code (https://github.com/betts-lab/asapseq-hiv-art).

### Initial ADT data processing

Sequencing reads were converted to fastq files and demultiplexed using bcl2fastq (Illumina). Reads were then inputted into kallisto bustools[76] to generate cell barcode by feature matrices. These matrices were loaded into R and checked for empty droplets (that is, background noise) using the emptyDrops function from the DropletUtils[77] package with a lower bound of 500 unique molecular identifiers (UMIs) for the in vitro/chronic datasets and 250 UMIs for the ART-treated dataset. Barcodes with FDRs < 0.01 were kept as genuine cells. This filtered matrix was then loaded into Seurat (v.4.1.1)[78] for centered log ratio transformation with a scale factor of 10,000. Harmony[79] was used for batch correction if several 10x GEM wells were used for the same individual. A Wilcoxon test (right-tailed) was used to select for features with a count distribution that was significantly different than background signal detected from isotype control antibodies, as analyzed similarly in a previous report[24]. A feature was not included in downstream analyses if P > 0.05 in at least four isotype controls.

### Identification of HIV+ cells

We built a custom Python pipeline (hiv-haystack; https://github.com/betts-lab/hiv-haystack), built and derived from epiVIA[22], to identify proviral reads from the BAM output file of cellranger-atac count. BAM records are parsed and selected into three groups: (1) both mates aligning to provirus, (2) both mates in host with a soft align clip and (3) one mate in provirus and one mate in host. In scenario 1, the original alignments were kept as bona fide proviral alignments. If there was a soft clip present, we aligned it to the host genome using bwa mem as done in epiVIA to assess whether an integration site is present. In scenario 2, any host soft clips were checked for exact matching to LTR sequences to allow for integration site detection. This was allowed only if a LTR alignment was reported (checked with BLAST against HXB2 5′ LTR). In scenario 3, the proviral mate was saved as a proviral fragment. If either mate had a soft clip, integration site checking was performed similar to scenarios 1 and 2. In all scenarios, only exact integration sites with exact nucleotide position are kept for downstream analyses. Our program also differs from epiVIA in that hiv-haystack allows for alignments to multiple sequences (from NFL genomes and single genome sequencing (SGS) of HIV Env), which enhances the detection of donor-derived sequences.

### Downstream data processing

After initial ATAC and ADT processing, barcodes that passed quality control metrics in both modalities were selected for downstream analyses. Clusters were annotated from a base panel of ADT expression patterns, imputed gene scores from ATAC, and differentially expressed markers present in any given cluster over all other cells. UMAP embeddings in lower dimension were generated using ArchR's latent semantic indexing implementation. Manual annotations were then used to subset cells for downstream ADT differential expression analysis between HIV+ and HIV− cells using Seurat FindMarkers (using 'DESeq2' (Wald test) or 'wilcox' (two-sided Wilcoxon rank sum) for the findMarkerMethod parameter) to limit false discoveries[80] and the subset of features that were significantly different from background expression. All P values from Seurat's FindMarkers function have been corrected for multiple comparisons using the Bonferroni method. For downstream ATAC differential expression analysis, manual clusters were grouped based on CD4+/− and HIV+/− (and by memory phenotype for the in vitro experiment) and pseudobulk replicates were generated. Clusters with less than 0.5% of the population were filtered out due to annotation difficulty. Peaks were called using the default 501-bp fixed-width method with ArchR and MACS2 (ref. [81]). Differential peaks were assessed using ArchR's getMarkerFeatures (which implements a two-sided Wilcoxon rank sum test and corrected for multiple tests using the Benjamini−Hochberg method) with bias parameters set to 'c(TSSEnrichment', 'log10(nFrags'))' as recommended by ArchR. Motif annotations of differential peaks were added using addMotifAnnotations from the CISBP dataset. For experiments with lower numbers of HIV+ cells, we used ArchR's implementation of chromVAR to determine differential motifs. getMarkerFeatures was called to determine differentially expressed motifs using the chromVAR deviation scores (that is, z score).

### Supervised machine learning

Samples were split into a 70% training and 30% testing dataset using the caTools package in R. Logistic regression models were developed using

the glm function with parameters 'family = binomial(link = 'logit').' Naive Bayesian models were constructed using the naivebayes package in R. RF models were constructed using the RF package in R with ntree = 4,000. Classifier performances (true positive rate and false positive rate) were calculated using the ROCR package in R and plotted using ggplot2.

### NFL SGS of proviral DNA from resting CD4[+] T cells

DNA was extracted from multiples of 4 million resting CD4[+] T cells according to the manufacturer's instructions (QIAamp DNA Mini and Blood Mini Kit, Qiagen). Amplification of NFL genomes was performed by limiting dilution, nested PCR using Platinum Taq High Fidelity Polymerase (Life Technologies, Thermo Fisher Scientific), adapted for NFL genomes with primary forward and reverse primers 5′-AAATCTCTAGCAGTGGCGCCCGAACAG-3′ and 5′-TGAGGGATCTCTAGTTACCAGAGTC-3′, respectively, followed by nested forward and reverse primers 5′-GCGGAGGCTAGAAGGAGAGAGATGG-3′ and 5′-GCACTCAAGGCAAGCTTTATTGAGGCTTA-3′, respectively[39]. Amplicons of appropriate size were sequenced directly on the Illumina MiSeq platform, inspected for evidence of priming from multiple templates, stop codons, large deletions or introduction of PCR error in early cycles; a threshold of 85% identity at each nucleotide position was used.

### HIV custom sequence annotation

HIV custom sequences from untreated and treated PLWH were loaded into the Gene Cutter tool (HIV LANL Database) for alignment to HIV genes/regions. The output was then loaded into our custom tool gene-CutterParser (https://github.com/wuv21/geneCutterParser) to extract the coordinate locations for any alignments. The coordinate locations were then used for alignment graphs in the Extended Data figures.

### Graphics

All figures were made in R with the following packages: gridExtra, ggplot2 (ref. [82]), ArchR and patchwork. All code to produce graphs can be found in the study GitHub repository.

### Reporting summary

Further information on research design is available in the Nature Portfolio Reporting Summary linked to this article.

### Data availability

Raw fastq files and processed cellranger-atac files are deposited in the NCBI Gene Expression Omnibus (GEO) under accession number GSE199727. Additional data for consensus alignments are provided in Supplementary File 1.

### Code availability

All code for downstream analysis is available at https://github.com/betts-lab/asapseq-hiv-art.

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

### Acknowledgements

We thank the individuals who donated samples to our study. We also thank members of the Betts, Vella and Bar laboratories for their assistance, A. Wells and J. Pippin for assistance with NovaSeq sequencing, the University of Pennsylvania Next-Generation Sequencing Core, University of Pennsylvania Flow Cytometry Core and the University of Pennsylvania Human Immunology Core. Additional thanks to W. Wang, G. Vahedi and B. Keele for helpful discussions regarding proviral alignment and software. Support for this study was provided by the following National Institutes of Health (NIH) grants: U19-A1-149680-02 (M.R.B.), R21-AI172629 (M.R.B), P01-AI31338 (M.R.B. and K.J.B.), K08-AI136660 (L.A.V.), T32-AI007632 (V.H.W.), P30-AIO45008 (Penn Center for AIDS Research) (M.R.B., L.A.V., K.J.B., P.T. and L.J.M.), UM-1AI164570 (BEAT-HIV Collaboratory), which is cosupported by the National Institute of Allergies and Infectious Diseases (NIAID), the National Institute of Mental Health (NIMH), the National Institute of Neurological Disorders and Stroke (NINDS), the National Institute on Drug Abuse (NIDA) and the Robert I. Jacobs Fund of The Philadelphia Foundation (M.R.B., K.J.B., P.T., L.J.M.). L.J.M. is also supported by the Herbert Kean, M.D., Family Professorship. CIENI-INER is supported by the Mexican Government (Programa Presupuestal PO16; Anexo 13 del Decreto del Presupuesto de Egresos de la Federación).

### Author contributions

V.H.W., K.J.B., L.A.V. and M.R.B. conceptualized experiments. P.M.d.R.E., F.T.-R., M.G.-N., Y.A.L.-V., S.A.-R., G.R.-T., P.T., L.J.M. and K.J.B. recruited and organized sample collection and cell preparations that were used for the study. V.H.W. and J.M.L.N. performed the ASAPseq and flow experiments. J.J., F.M. and K.J.B. conducted and compiled the consensus viral sequencing. V.H.W. and S.N. performed the ASAPseq data analysis. V.H.W., L.A.V. and M.R.B. wrote the manuscript.

### Competing interests

M.R.B. is a consultant for Interius BioTherapeutics. No other conflicts are reported by the authors.

### Additional information

**Extended data** is available for this paper at https://doi.org/10.1038/s41590-022-01371-3.

**Correspondence and requests for materials** should be addressed to Laura A. Vella or Michael R. Betts.

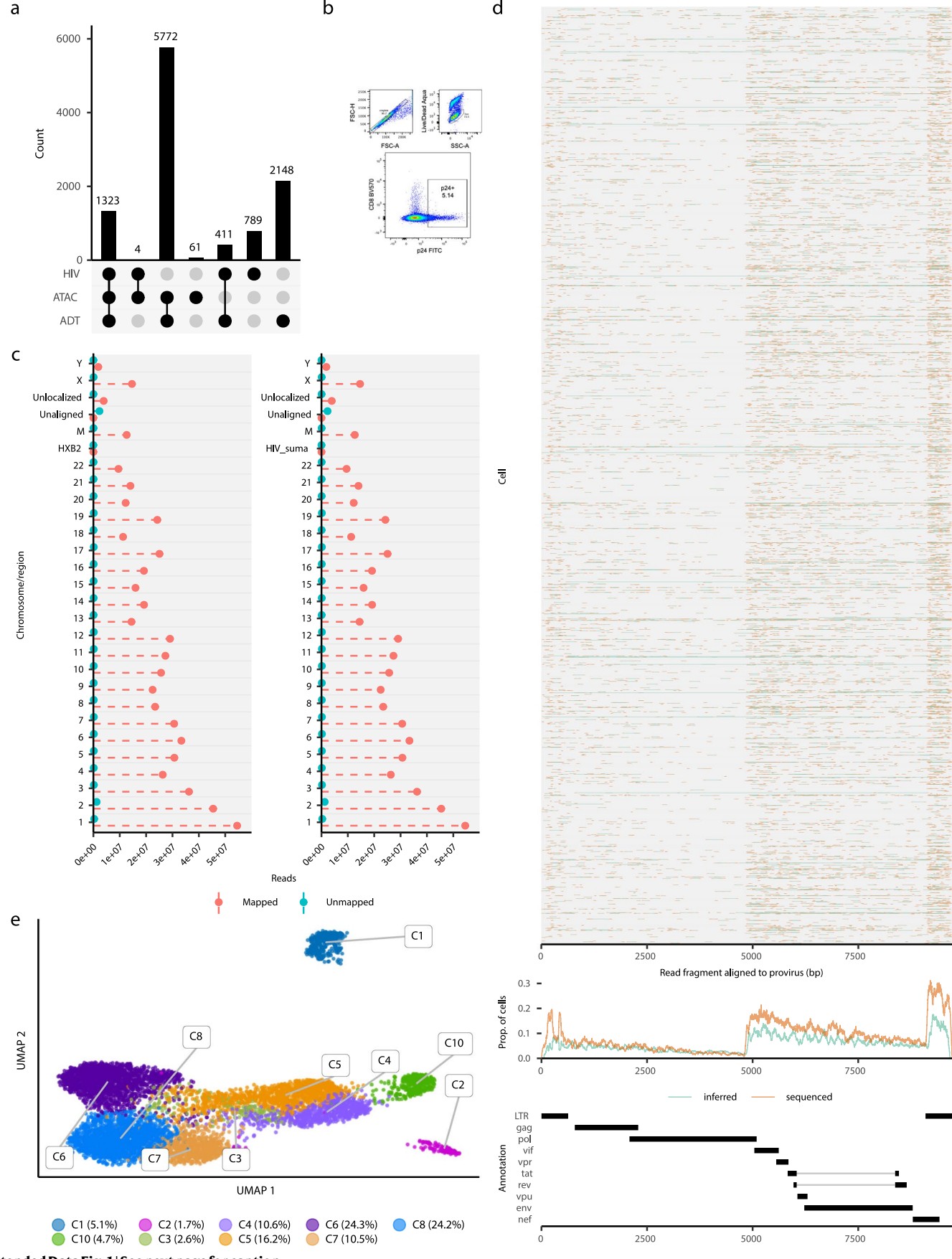

**Extended Data Fig. 1 | See next page for caption.**

**Extended Data Fig. 1 | Properties of ASAPseq library for in vitro model.** (A) UpSet plot of unique cell barcodes that were collected from each modality (ATAC versus ADT) and whether or not the barcode was associated with proviral reads (HIV). Barcodes that passed ATAC and ADT quality checks (see Methods) were used for downstream analyses. (B) Flow cytometry plot of cell culture before conducting ASAPseq analysis. Top two plots (from left to right) indicate gating strategy. Value in the highlighted box for the bottom plot is the percent of total live singlets that are p24 $^+$. (C) Reported mapped and unmapped read-segments by chromosome (as determined by samtools idxstats) from alignment of ASAPseq dataset of uninfected PBMC (Mimitou et al., 2021) to chimeric reference genomes with HXB2 (left) or SUMA (right). HIV genomes were added as a separate chromosome during creation of the chimeric reference genome. (D) (top) Sequenced regions that are aligned by bwa mem to the proviral genome (SUMA) and recovered by hiv-haystack. Each row is a cell and each column is a base pair spanning the proviral genome. Regions in orange indicate actual reported coverage while regions in blue indicate inferred coverage if provirus was intact as paired-end sequencing can only obtain at most 50 bp from either end of the genomic/transposed fragment if the genomic fragment is > 50 bp. Many LTR alignments can be ambiguous and it is unclear whether the actual read is in the 3′ LTR or 5′ LTR. The primary alignment from bwa-mem is recorded here. (middle) Proportion of coverage is reported across all cells spanning the entire proviral genome. (bottom) Genome map of SUMA. (E) UMAP representation of the ATAC component with numeric labeling prior to manual annotation.

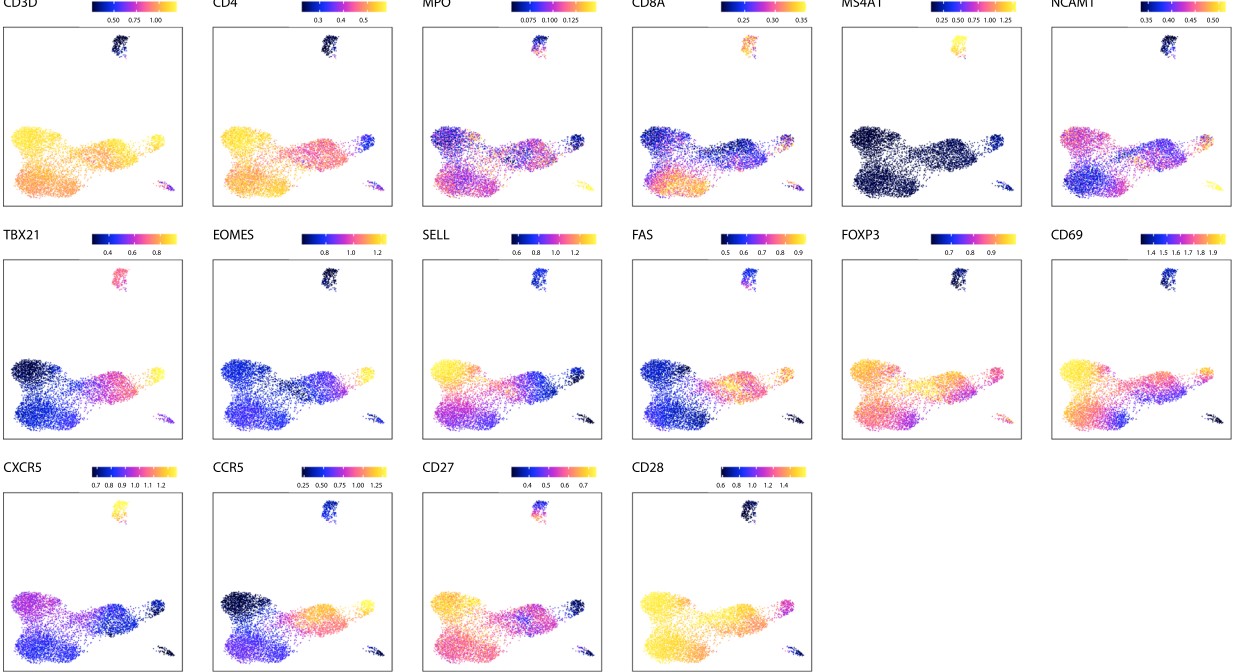

**Extended Data Fig. 2 | Cluster annotation panels for in vitro model.** (A) Each subplot shows the ADT signal for a specific surface antigen for each cluster as seen in Extended Data Fig. 1E. X-axis values are normalized count values as processed via Seurat. (B) Each subplot shows the imputed gene activity score overlaid on the UMAP coordinate space as seen in Extended Data Fig. 1E. Gene activity score was calculated by ArchR and imputed using MAGIC to aid in visual interpretation as recommended by ArchR. Color scale indicates log2(normalized counts +1).

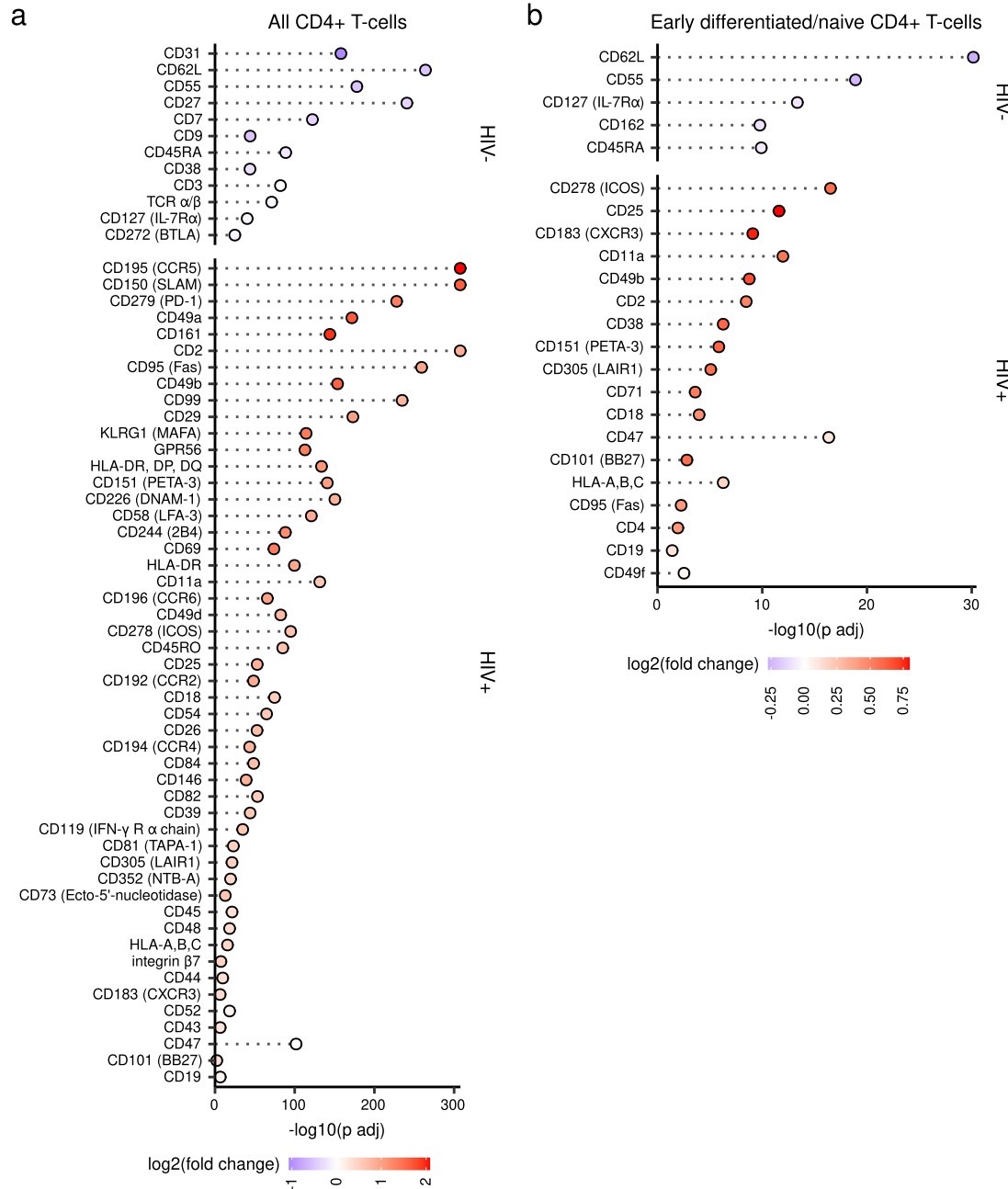

**Extended Data Fig. 3 | Differential expression of select antigens for in vitro model.** Differential expression of surface antigens was assessed (DESeq2 method in Seurat; two-sided with multiple comparison adjustment using Bonferroni-Hochberg method) to compare between HIV- or HIV + cells in specific cell groupings: (A) all CD4 + T-cells (n = 1279 cells for HIV + and n = 5315 cells for HIV-) and (B) early differentiated CD4 + T-cells (n = 216 cells for HIV + and n = 3958 cells for HIV-). Markers are ranked in (A-B) by π-score (see Fig. 1E legend). All markers shown have an adjusted p-value < 0.05.

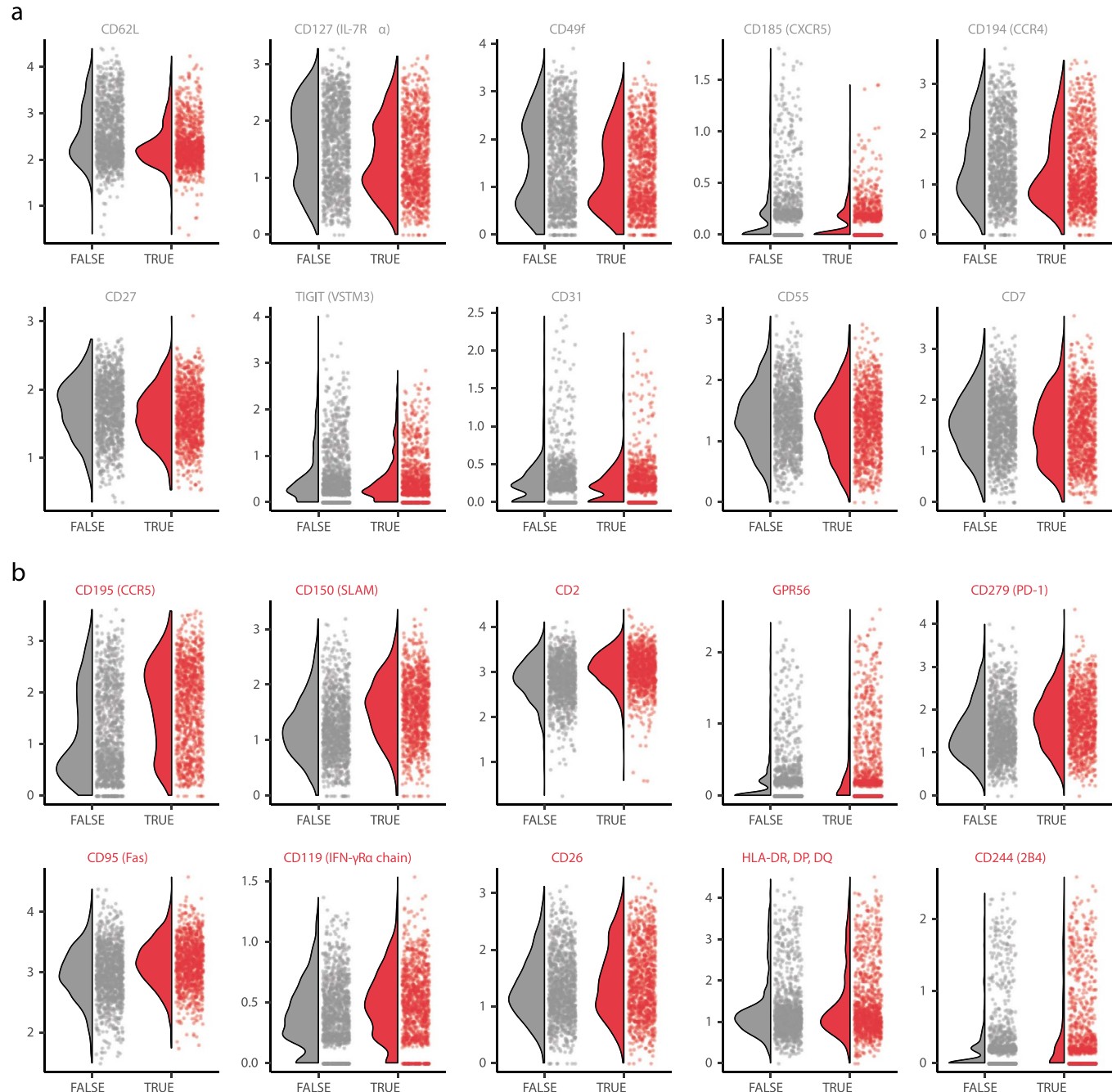

**Extended Data Fig. 4 | Differential expression of select antigens in activated T-cells for in vitro model.** Violin-scatter plots are shown for the top 10 surface markers from Fig. 1E that are enriched in (A) activated HIV- cells and (B) activated HIV + cells. FALSE indicates HIV- cells while TRUE indicates HIV + cells. Markers are ordered from left to right; top to bottom by order of decreasing |π-score| as seen in Fig. 1E.

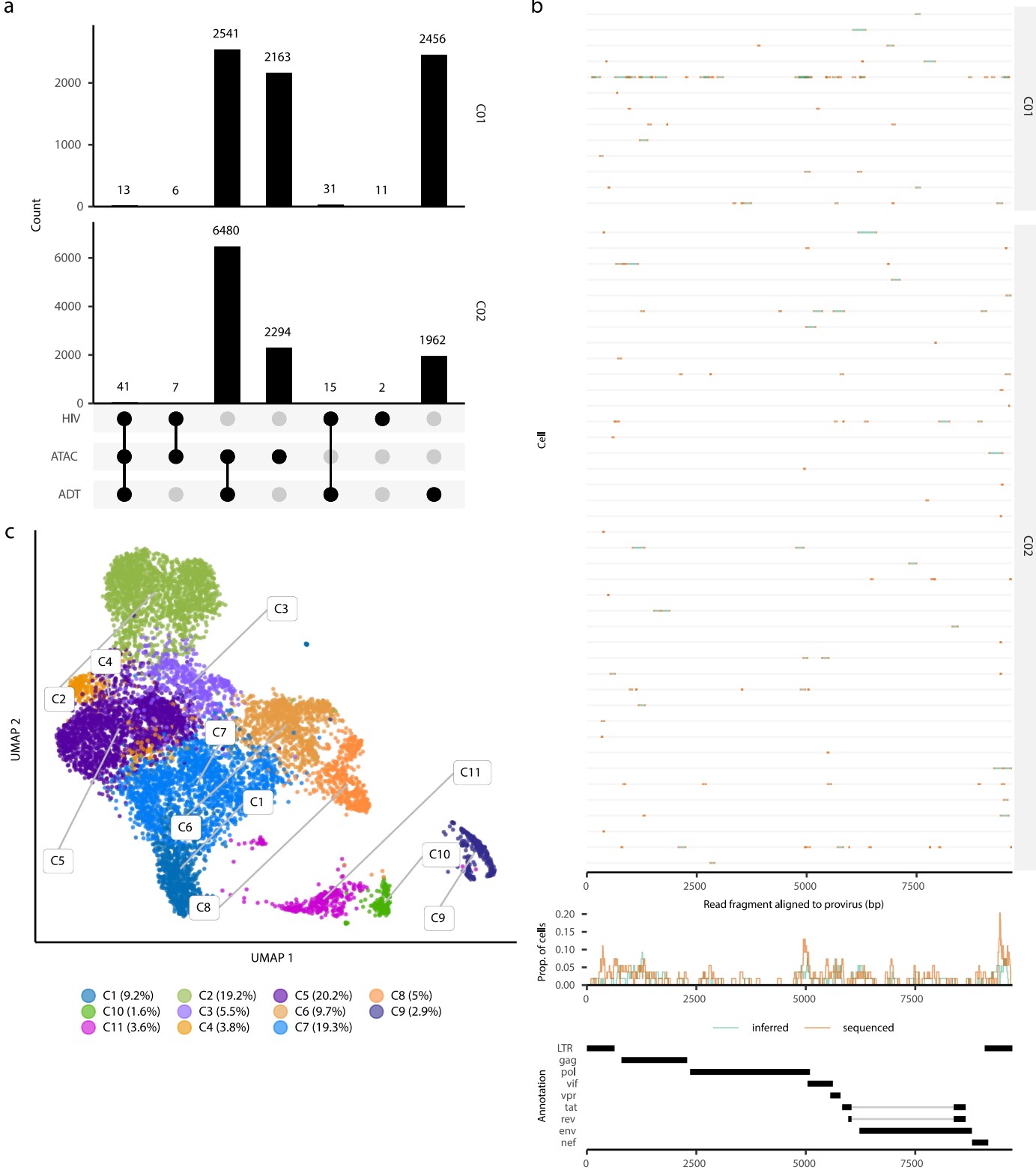

**Extended Data Fig. 5 | Properties of ASAPseq library during chronic infection.** (A) UpSet plot of unique cell barcodes that were collected from each modality (ATAC versus ADT) and detection of proviral reads (HIV) separated by individual. (B) (top) Sequenced regions that are aligned by bwa mem to the proviral genome (HXB2) and recovered by hiv-haystack. Each row is a cell and each column is a base pair spanning the proviral genome. Refer to Extended Data Fig. 1 legend for more detailed information. (C) UMAP representation of the ATAC component with numeric labeling prior to manual annotation.

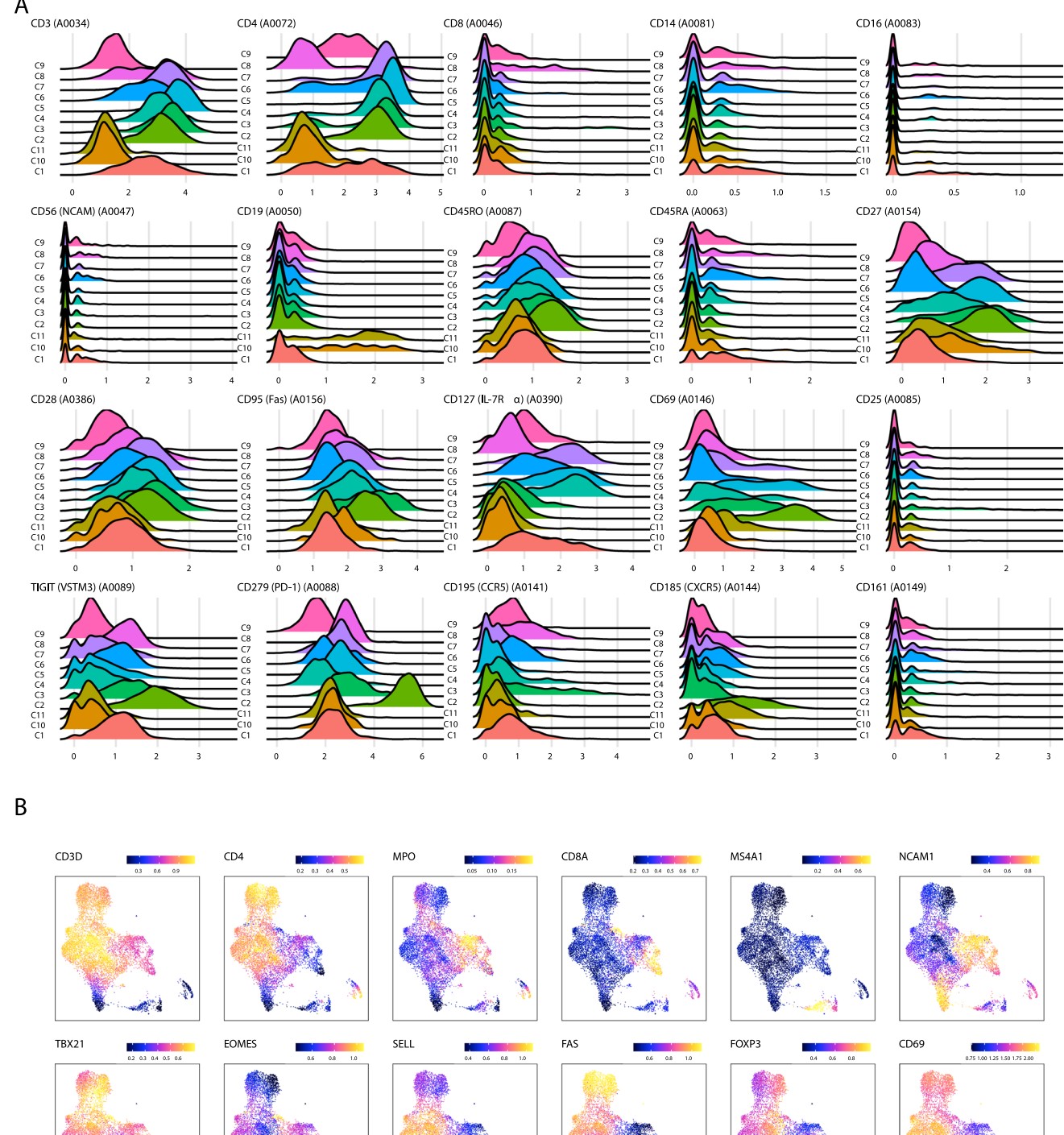

**Extended Data Fig. 6 | Cluster annotation panels during chronic infection.** (A) Each subplot shows the ADT signal for a specific surface antigen for each cluster as seen in Extended Data Fig. 5C. X-axis values are normalized count values as processed via Seurat. (B) Each subplot shows the imputed gene activity score overlaid on the UMAP coordinate space as seen in Extended Data Fig. 5C. Gene activity score was calculated by ArchR and imputed using MAGIC to aid in visual interpretation as recommended by ArchR.

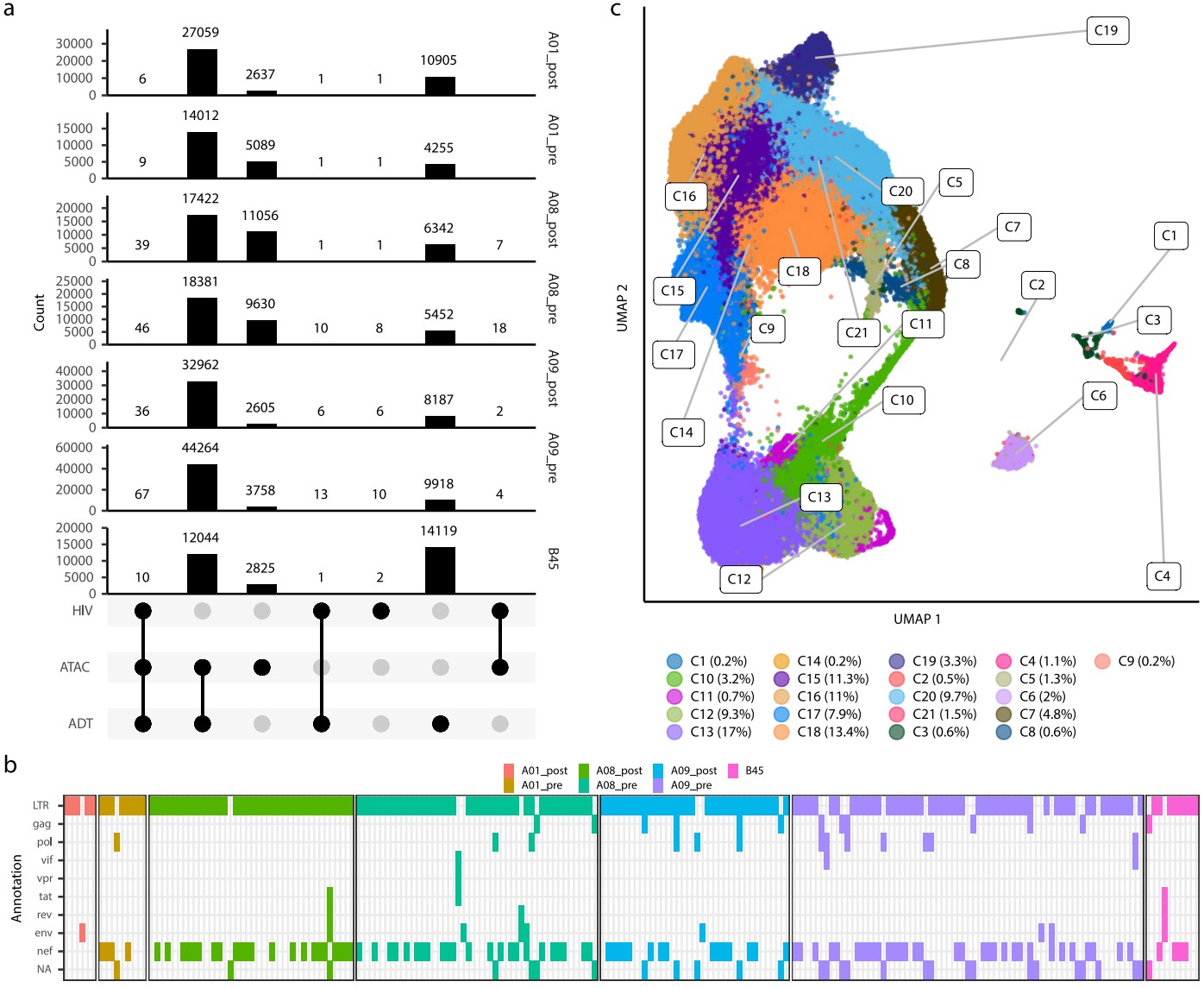

**Extended Data Fig. 7 | Properties of ASAPseq library during treated infection.**
(A) UpSet plot of unique cell barcodes that were collected from each modality (ATAC versus ADT) and detection of proviral reads (HIV) separated by individual. (B) Sequenced regions that are aligned by bwa mem to the proviral genome (autologous + HXB2) and recovered by hiv-haystack. Each column represents an unique infected cell, separated by the individual. The annotated HIV genomic region (as determined from Gene Cutter) is displayed per infected cell. (C) UMAP representation of the ATAC component with numeric labeling prior to manual annotation.

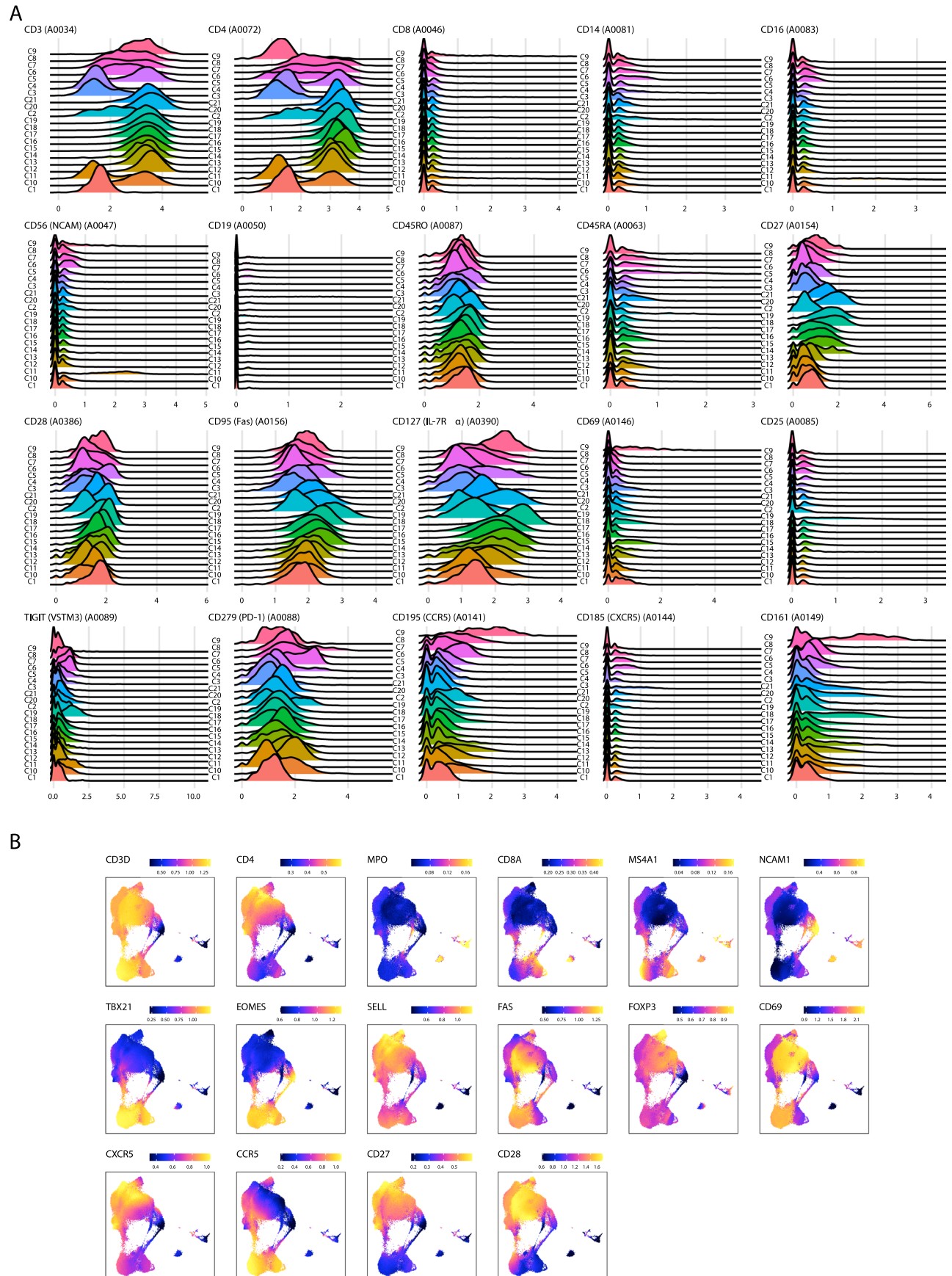

**Extended Data Fig. 8 | Cluster annotation panels during treated infection.** (A) Each subplot shows the ADT signal for a specific surface antigen for each cluster as seen in Extended Data Figure 7C. X-axis values are normalized count values as processed via Seurat. (B) Each subplot shows the imputed gene activity score overlaid on the UMAP coordinate space as seen in Extended Data Figure 7C. Gene activity score was calculated by ArchR and imputed using MAGIC to aid in visual interpretation as recommended by ArchR.

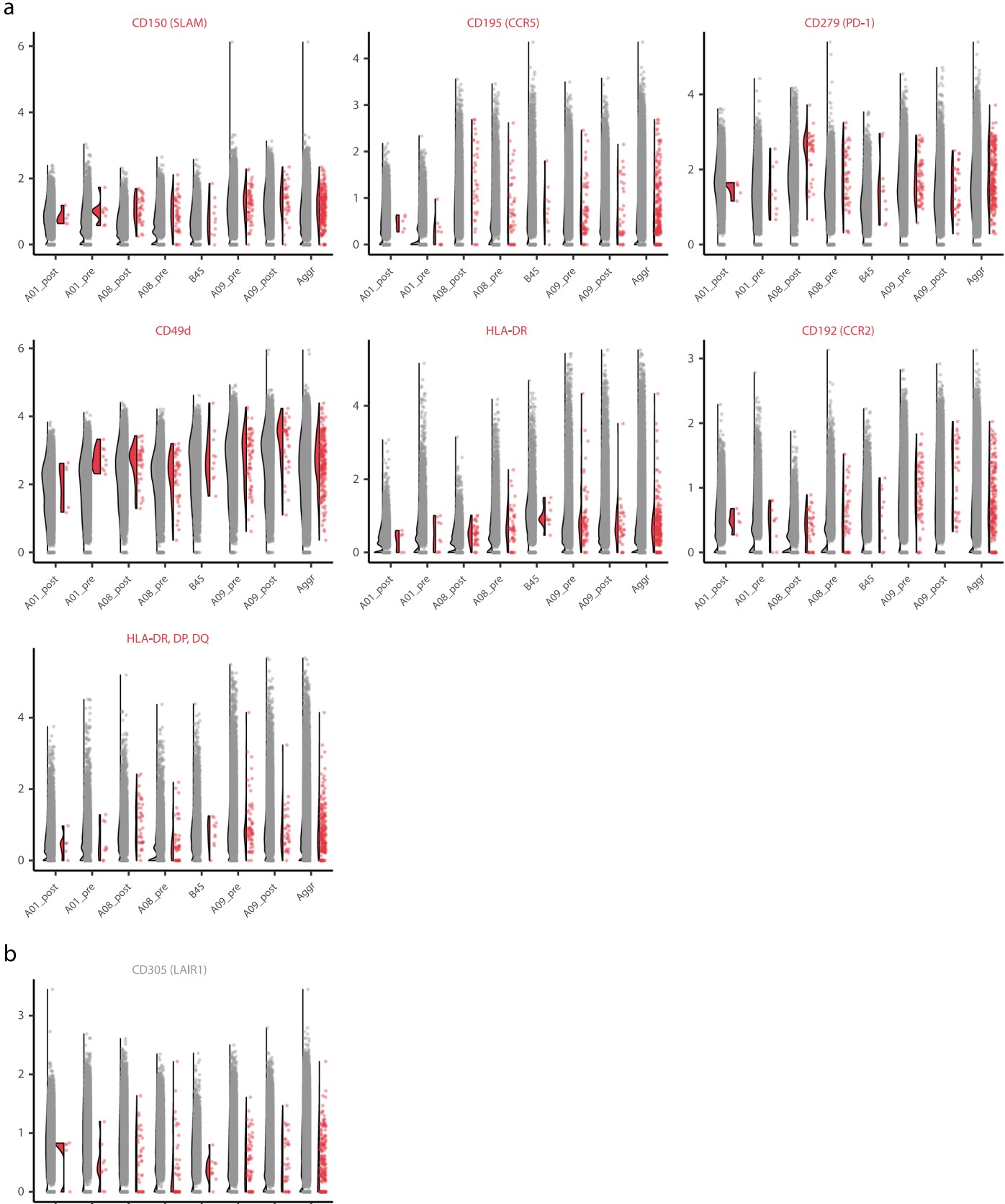

**Extended Data Fig. 9 | See next page for caption.**

**Extended Data Fig. 9 | Differential expression of select antigens during treated infection.** Violin-scatter plots are shown for all significantly expressed (adjusted p value < 0.05; two-sided Wilcoxon with multiple comparison adjustment using Bonferroni method) surface markers (see Fig. 5B for significance and fold change values) that are enriched in (A) all CD4 + HIV + T-cells and (B) all CD4 + HIV- T-cells. FALSE indicates HIV- cells while TRUE indicates HIV + cells. Markers are ordered from left to right; top to bottom by order of decreasing |π-score| as seen in Fig. 5B. Cells are separated by individual and the aggregate data (across individuals) is also shown.

# Reporting Summary

## Statistics

For all statistical analyses, confirm that the following items are present in the figure legend, table legend, main text, or Methods section.

| n/a | Confirmed | |
|---|---|---|
| ☐ | ☒ | The exact sample size (*n*) for each experimental group/condition, given as a discrete number and unit of measurement |
| ☐ | ☒ | A statement on whether measurements were taken from distinct samples or whether the same sample was measured repeatedly |
| ☐ | ☒ | The statistical test(s) used AND whether they are one- or two-sided<br>*Only common tests should be described solely by name; describe more complex techniques in the Methods section.* |
| ☒ | ☐ | A description of all covariates tested |
| ☐ | ☒ | A description of any assumptions or corrections, such as tests of normality and adjustment for multiple comparisons |
| ☐ | ☒ | A full description of the statistical parameters including central tendency (e.g. means) or other basic estimates (e.g. regression coefficient) AND variation (e.g. standard deviation) or associated estimates of uncertainty (e.g. confidence intervals) |
| ☒ | ☐ | For null hypothesis testing, the test statistic (e.g. *F*, *t*, *r*) with confidence intervals, effect sizes, degrees of freedom and *P* value noted<br>*Give P values as exact values whenever suitable.* |
| ☒ | ☐ | For Bayesian analysis, information on the choice of priors and Markov chain Monte Carlo settings |
| ☒ | ☐ | For hierarchical and complex designs, identification of the appropriate level for tests and full reporting of outcomes |
| ☒ | ☐ | Estimates of effect sizes (e.g. Cohen's *d*, Pearson's *r*), indicating how they were calculated |

*Our web collection on statistics for biologists contains articles on many of the points above.*

## Software and code

Policy information about availability of computer code

| Data collection | cellranger-atac (v2.0.0); index-hopping-filter (v1.1); kallisto (v0.46.2); bustools (v0.40.0); AMULET (v1.1); ArchR (v1.0.2); Seurat (v4.1.1); chromVAR (v1.16.0); DropletUtils (v1.14.2); DESeq2 (v1.34.0); naivebayes (v0.9.7); caTools(v1.18.2), ROCR (v1.0-11); randomForest (v4.7-1.1); hiv-haystack (v1; https://github.com/betts-lab/hiv-haystack); Gene Cutter (HIV LANL Database; web version; https://www.hiv.lanl.gov/content/sequence/GENE_CUTTER/cutter.html); geneCutterParser (v1; https://github.com/wuv21/geneCutterParser); R (v4.1.1); gridExtra (v2.3); ggplot2 (v3.3.6); patchwork (v1.1.1) |
|---|---|
| Data analysis | Custom code is available at https://github.com/betts-lab/asapseq-hiv-art |

For manuscripts utilizing custom algorithms or software that are central to the research but not yet described in published literature, software must be made available to editors and reviewers. We strongly encourage code deposition in a community repository (e.g. GitHub). See the Nature Portfolio guidelines for submitting code & software for further information.

## Data

Policy information about availability of data

All manuscripts must include a data availability statement. This statement should provide the following information, where applicable:
- Accession codes, unique identifiers, or web links for publicly available datasets
- A description of any restrictions on data availability
- For clinical datasets or third party data, please ensure that the statement adheres to our policy

Raw fastq files and processed cellranger-atac files are deposited in the NCBI Gene Expression Omnibus (GEO) under series accession number GSE199727. CISBP human transcription factor motif data (http://cisbp.ccbr.utoronto.ca/) were included in the ArchR pacakge by the original developers. HIV LANL database (https://www.hiv.lanl.gov/content/sequence/GENE_CUTTER/cutter.html) was used to access the Gene Cutter tool.

## Human research participants

Policy information about studies involving human research participants and Sex and Gender in Research.

| | |
|---|---|
| Reporting on sex and gender | All untreated donors are male as reported by Centro de Investigación en Enfermedades Infecciosas. All treated donors are male as previously reported (Bar et al., 2016) and as reported for samples provided by the University of Pennsylvania Human Immunology Core and the BEAT-HIV program cohort. |
| Population characteristics | Age was not reported at the participant level for the original study (Bar et al., 2016). However, the age range was reported as between 34 to 52 years old. Donor ages for PLWH with chronic HIV infection are reported in Table 1. For the ex vivo data, all participants had diagnosis of HIV-1 infection and all had suppressed viral load under ART treatment at the time of sample collection. No genotypic information on participants was available for this study. Please refer to Table 1 for more information on all donors included in this study. |
| Recruitment | No participants were recruited specifically for this study. ART-treated samples were provided by Drs. Katie Bar, Pablo Tebas, and Luis Montaner from previously published studies. Chronic HIV samples were provided by Drs. del Rio Estrada, Torres-Ruiz, Gonzales-Navarro, Luna-Villalobos, Avila-Rios, and Reyes-Teran. |
| Ethics oversight | This study was approved by the Institutional Review Boards at the University of Pennsylvania and the University of Alabama at Birmingham. This study complies with all relevant ethical regulations. Persons living with chronic HIV (n = 2) were originally recruited by the Centro de Investigación en Enfermedades Infecciosas at the Instituto Nacional de Enfermedades Respiratorias (CIENI-INER) in Mexico City, Mexico. All donors provided informed consent for lymph node tissue donation in compliance with protocols set forth by the Ethics Committee and the Ethics in Research Committee of the INER (study number: B03-16) and the Institutional Review Board at the University of Pennsylvania (Philadelphia, PA). ART-treated samples (n = 3), A01, A08, and A09, were provided from the ACTG clinical trial A5340, which was conducted with protocols set forth by the Institutional Review Boards at the University of Pennsylvania and the University of Alabama at Birmingham and was previously published (Bar et al., 2016). The original clinical trial included provisions for research related to this study. All donors for this study provided informed consent in compliance with protocols set forth by the respective institutional review boards. Another ART-treated PLWH (n = 1), B45, was recruited from the BEAT-HIV program cohort where an apheresis sample was collected under ART. All participants were compensated for their time and study visits. No additional compensation was provided for this study. |

Note that full information on the approval of the study protocol must also be provided in the manuscript.

# Field-specific reporting

Please select the one below that is the best fit for your research. If you are not sure, read the appropriate sections before making your selection.

☒ Life sciences          ☐ Behavioural & social sciences          ☐ Ecological, evolutionary & environmental sciences

For a reference copy of the document with all sections, see nature.com/documents/nr-reporting-summary-flat.pdf

# Life sciences study design

All studies must disclose on these points even when the disclosure is negative.

| | |
|---|---|
| Sample size | 2 separate ex vivo untreated samples (C01 and C02) were analyzed. 7 separate ex vivo treated samples were analyzed (A01 (two timepoints), A08 (two timepoints), A09 (two timepoints), and B45). No sample size calculations were performed as the samples were provided based on availability and rarity of infected cells. |
| Data exclusions | No datasets were excluded. Filtering of cells was done for each dataset for potential index hopping (if applicable), quality control, potential multiplets, and small (<0.5% of total number of QC passing cells) clusters due to the inability to annotate properly. A wilcoxon test (right-tailed) was used to select for surface antigen features with a count distribution that was significantly different than background signal |

detected from isotype control antibodies, as analyzed similarly in a prior report (Swanson et al., 2021). A feature was not included in downstream analyses if p > 0.05 in at least 4 isotype controls.

| | |
|---|---|
| Replication | No attempts at replication as these are direct ex vivo profiling of samples from individuals. In vitro experiment was not replicated as this was a proof of concept experiment and was confirmed for feasibility by the ex vivo experiments. Pseudo-bulk replicates were generated as part of the ArchR package in downstream differential analyses due to sparse single-cell epigenetic datasets (refer to ArchR documentation). |
| Randomization | This study was not designed as a clinical trial and all individuals were PLWH. The primary study measure was between cells within the same individual. Therefore, no randomization or blinding was performed in the study design or analysis. |
| Blinding | Similar to the above rationale, this study was not designed as a clinical trial and all individuals were PLWH. The primary study measure was between cells within the same individual. Therefore, no randomization or blinding was performed in the study design or analysis. |

# Reporting for specific materials, systems and methods

We require information from authors about some types of materials, experimental systems and methods used in many studies. Here, indicate whether each material, system or method listed is relevant to your study. If you are not sure if a list item applies to your research, read the appropriate section before selecting a response.

## Materials & experimental systems

| n/a | Involved in the study |
|---|---|
| ☐ | ☒ Antibodies |
| ☒ | ☐ Eukaryotic cell lines |
| ☒ | ☐ Palaeontology and archaeology |
| ☒ | ☐ Animals and other organisms |
| ☒ | ☐ Clinical data |
| ☒ | ☐ Dual use research of concern |

## Methods

| n/a | Involved in the study |
|---|---|
| ☒ | ☐ ChIP-seq |
| ☐ | ☒ Flow cytometry |
| ☒ | ☐ MRI-based neuroimaging |

## Antibodies

| | |
|---|---|
| Antibodies used | All antibodies used are commercially available and are described in the Methods and in Supplemental Table 1.<br><br>For proteogenomics:<br>- TotalSeqA Human Universal Cocktail, V1.0 (BioLegend #399907)<br><br>For flow cytometry:<br>- CD8 BV570 (BioLegend #301038, clone RPA-T8)<br>- p24 FITC (Beckman Coulter #6604665; clone KC57) |
| Validation | All antibodies are published clones and validated by the vendors. For proteogenomic applications, the TotalSeqA Human Universal Cocktail (v1.0) was validated by BioLegend for proteogenomic applications and human reactivity. For flow cytometry, the CD8 BV570 antibody was validated by BioLegend for flow cytometry applications and human reactivity while the p24 FITC clone was validated by Beckman Coulter for flow cytometry applications and reactivity to HIV p24. |

## Flow Cytometry

### Plots

Confirm that:

☒ The axis labels state the marker and fluorochrome used (e.g. CD4-FITC).

☒ The axis scales are clearly visible. Include numbers along axes only for bottom left plot of group (a 'group' is an analysis of identical markers).

☒ All plots are contour plots with outliers or pseudocolor plots.

☒ A numerical value for number of cells or percentage (with statistics) is provided.

### Methodology

| | |
|---|---|
| Sample preparation | Peripheral blood mononuclear cells (PBMCs) were obtained from an HIV-negative donor apheresis from the Human Immunology Core (HIC) at the University of Pennsylvania. Bulk CD4+ T cells were negatively enriched by bead separation and infected with HIV-1. Staining and flow cytometry were based on a previously published protocol (Kuri-Cervantes et al., 2020). Approximately 1.5 million cells from the in vitro infection culture were spun down at 400 x g for 5 min and resuspended in 45μl of PBS. Live/dead staining was performed using 5μl of a 1:60 dilution stock of prepared Live/Dead Fixable Aqua Dead Cell Stain (Invitrogen). Cells were stained for 5 minutes in the dark at room temperature. A staining cocktail with FACS buffer and CD8 BV570 (BioLegend #301038, clone RPA-T8) was added for a 10 minute stain in the dark at room temperature. 1ml FACS buffer was added and the cells were spun down at 400 x g for 5 min. Cells were permeabilized with 250μl of BD Cytofix/ |

Cytoperm solution (BD #554714) for 18 minutes in the dark at room temperature. 1ml of BD Perm/Wash Buffer (BD #554714) was added and cells were spun down at 600 x g for 5 min. After supernatant was discarded, cells were resuspended in staining solution containing anti-p24 FITC (Beckman Coulter #6604665; clone KC57) and BD Perm/Wash Buffer for a final staining volume of 50µl. Cells were stained in the dark for 1 hour at room temperature. Cells were washed with 1ml of BD Perm/Wash Buffer and fixed with 350ul of 1% paraformaldehyde. 75345 events were acquired.

Instrument | BD FACS Symphony A5 cytometer

Software | FlowJo (v10.8.0)

Cell population abundance | Final population (p24+) was 5.10% (2546) of 49572 live singlet cells.

Gating strategy | Singlets (FSC-A by FSC-H) -> Live cells (negative staining for Live/Dead Aqua) -> p24+ CD8- T cells (p24+ and CD8-)

☒ Tick this box to confirm that a figure exemplifying the gating strategy is provided in the Supplementary Information.

