## [Peer Review File · Nature Immunology]

Peer Review Information

Journal: Nature Immunology

Manuscript Title: Profound phenotypic and epigenetic heterogeneity of the HIV-1 infected CD4+ T cell reservoir

Corresponding author name(s): Michael Betts

Reviewer Comments & Decisions:

Decision Letter, initial version:

24th Aug 2022

Dear Professor Betts,

As you know your Resource, "Profound phenotypic and epigenetic heterogeneity of the HIV-1 infected CD4+ T cell reservoir" has now been seen by 2 referees. You will see from their comments below that while they find your work of interest, some important points are raised. We are interested in the possibility of publishing your study in Nature Immunology.

We therefore invite you to revise your manuscript as described in your recent email. We do request you include the LN data as a supplementary figure. Please highlight all changes in the manuscript text file in Microsoft Word format.

* If you have not done so already please begin to revise your manuscript so that it conforms to our Resource format instructions at <http://www.nature.com/ni/authors/index.html>. Refer also to any guidelines provided in this letter.

* Please include a revised version of any required reporting checklist. It will be available to referees to aid in their evaluation of the manuscript goes back for peer review. They are available here:

Reporting summary:

When submitting the revised version of your manuscript, please pay close attention to our [href="https://www.nature.com/nature-portfolio/editorial-policies/image-integrity">Digital Image Integrity Guidelines.](https://www.nature.com/nature-portfolio/editorial-policies/image-integrity) and to the following points below:

Please use the link below to submit your revised manuscript and related files: [REDACTED]

We hope to receive your revised manuscript within two weeks. If you cannot send it within this time, please let us know. We will be happy to consider your revision so long as nothing similar has been accepted for publication at Nature Immunology or published elsewhere.

Nature Immunology is committed to improving transparency in authorship. As part of our efforts in this direction, we are now requesting that all authors identified as 'corresponding author' on published papers create and link their Open Researcher and Contributor Identifier (ORCID) with their account on the Manuscript Tracking System (MTS), prior to acceptance. ORCID helps the scientific community achieve unambiguous attribution of all scholarly contributions. You can create and link your ORCID from the home page of the MTS by clicking on 'Modify my Springer Nature account'. For more information please visit www.springernature.com/orcid.

Sincerely,

Jamie D.K. Wilson, D.Phil
Chief Editor
Nature Immunology
212 726 9207
j.wilson@us.nature.com

Reviewers' Comments:

Reviewer #1:

Remarks to the Author:

BRIEF SUMMARY OF THE MANUSCRIPT

In this manuscript, Wu and colleagues developed ASAP-Seq, a novel methodology that aligns autologous HIV viral sequences with ATAC-Seq (measuring chromatin availability) and CITE-Seq (protein expression). By combining these methodologies, the authors have gained unprecedented insight into how heterogeneous CD4+ T cells harboring HIV-1 proviral DNA are in patients undergoing antiretroviral treatment. This work supports the notion that therapeutic strategies to eradicate HIV viral reservoirs might need to be adapted to patient-specific epigenetic and cellular features that could be uncovered by this platform.

NOVELTY: High

DATA, METHODOLOGY ETC

With ongoing efforts to develop therapeutic strategies to eradicate HIV viral reservoirs, technologies

with the capability to provide viral, epigenetic, and cellular information simultaneously are urgently needed. It is known that CD4+ T cells are the main source of HIV-1 provirus DNA but defining a signatures of these cells is still challenging due to their scarcity in blood.

A considerable strength of this manuscript is the thorough validation of the platform. By using a model of in vitro infection of primary CD4+ T cells from an HIV-1 seronegative individual using a lab-adapted HIV with a known sequence, the authors elegantly identified epigenetic and cellular changes in HIV-1+ CD4+ T cells, which they used to build up a regression model that identifies with high accuracy several markers that predict infection. Another crucial part of the work that is extremely valuable was the use of autologous patient-derived HIV sequences, covering for the high mutational rate detected in the proviral sequence.

LIMITATIONS

1. Limited cell numbers (~150 HIV-1+ CD4+ T cells)
2. Requirement of the provirus to be in accessible chromatin regions, potentially biasing a towards a subset of latent reservoirs, albeit that most likely to contribute to ongoing clinical problems. Also, the methods could not differentiate between competent and defective viruses.

Despite the technical limitations and the lack of identification of a "universal" signature of HIV-1+ cells, this manuscript provides a valuable technology, robust computational pipelines, and the creation of an atlas of HIV-1 infected cells that could be used and further curated by other HIV researchers, with the expectation in increasing the number of sequenced cells.

SPECIFIC COMMENTS

1. HIV viral reservoirs are predominantly established in lymph nodes and the gut. Have the authors considered using LN samples to increase the number of cells harbouring HIV proviral DNA and ask whether or not the profiles of these cells are similar?
2. Three of the donors in the study previously experienced therapy interruption (enrolled in the VRC01 A5340 trial) but were under at least six months (for the post-ATI timepoint) of ART suppression at the time of analysis. It is unclear how long the subjects had been on ART before enrolment in the trial. Could the authors provide information on this and longitudinal viral load data, especially on the timepoint where ART was interrupted?
3. The authors include an impressive array of 154 CITE-Seq antibodies for protein analysis. Can they provide the basis for their selection?
4. In Table 1, information on HIV clade and ethnicity would be valuable for any future cell atlas, as proposed by the authors in the discussion.
5. It is striking that the authors found strong similarities in the phenotype of HIV-1+ CD4 T cell subsets before and after ART interruption. While possibly outside the scope of this manuscript, it would be interesting to dissect potential epigenetic changes that a temporal increase of viral load could have driven.
6. Figure 4 Panels A-D have incomplete labelling of the x-axes. Panel B could be more precise by plotting the $-\log_{10}$ p-value. Finally, in panel F the y axis needs clarification.

Reviewer #2:

Remarks to the Author:

The authors have developed a potentially important method for characterizing the HIV reservoir at the single cell level. They use a single cell ATAC-seq (scATACseq) approach to characterize individual cells that have reads containing HIV sequence. This is done first with cells infected in vitro with HIV and then with cells from persons living with HIV (PLWH) on antiretroviral therapy (ART). One goal seems to have been to identify surface markers unique to infected cells. The principal finding is that proviruses integrated into accessible chromosomal sites are present in a wide variety of different CD4 subpopulations.

The authors are to be congratulated for studying an important problem with a novel approach. This approach could make an important contribution to HIV cure research. Given the potential importance of the work, the authors should be encouraged to address the comments below to make the paper as strong as possible:

- 1) The paper is written in dense “genomics-speak” and is difficult to follow in places. For example, there is frequent use of the term “enrichment”, but the reader is left with little sense of the degree of enrichment from the statistical jargon that comprises much of the paper. Consider CD2 for example. This is a hit in some of the analyses but typically this protein shows a tight unimodal distribution on T cells. If there is biologically significant enrichment of markers like CD2 on infected cells, then it should be possible to test this experimentally by sorting CD2 high and low cells and measuring HIV.
- 2) Related to point 1, there is no attempt in the paper to experimentally validate any of the hits or relationships identified in the bioinformatic analysis.
- 3) Some aspects of the method need to be clarified. The distribution of reads throughout the genome shows a peak in the middle of the genome that is not explained. The detection of HIV in B cells and monocytes (Fig.1b) casts some doubt on the reliability of the method.
- 4) The study largely avoids key biological questions that the method could address, if used in combination with other assays. For example, what is the relationship between accessibility and viral gene expression?
- 5) It is unclear to what extent the findings contradict the large number of papers that claim that HIV is preferentially located in particular subsets? A better effort could be made to cite these papers and also papers that show wide distribution of HIV in different subsets.

6) It is likely that many of the proviruses detected in the assay are defective. This issue is only briefly mentioned, and it is unclear how this would affect the conclusions.

7) Similarly, it is likely that many of the proviruses detected are from expanded clones. It would be helpful to know how members of the same clone of HIV infected CD4+ T cells differ with regard to features measured in this assay.

8) The legends for the main and supplemental figures should be rewritten to clearly state what was tested in each panel and what the controls were.

Reviewer #3:

None

Author Rebuttal to Initial comments

Dear Dr. Wilson,

We would like to thank you and the reviewers for the combined time and effort in reviewing our manuscript. All reviewers recognized the valuable insights into HIV reservoir afforded by our manuscript along with the novelty of our methodology. We have taken additional care in the revised version to address reviewer concerns.

Our replies and revisions to the reviewers and their comments are described below in blue text.

Reviewer #1

BRIEF SUMMARY OF THE MANUSCRIPT

In this manuscript, Wu and colleagues developed ASAP-Seq, a novel methodology that aligns autologous HIV viral sequences with ATAC-Seq (measuring chromatin availability) and CITE-Seq (protein expression). By combining these methodologies, the authors have gained unprecedented insight into how heterogeneous CD4+ T cells harboring HIV-1 proviral DNA are in patients undergoing antiretroviral treatment. This work supports the notion that therapeutic strategies to eradicate HIV viral reservoirs might need to be adapted to patient-specific epigenetic and cellular features that could be uncovered by this platform.

NOVELTY: High

DATA, METHODOLOGY ETC

With ongoing efforts to develop therapeutic strategies to eradicate HIV viral reservoirs, technologies with the capability to provide viral, epigenetic, and cellular information simultaneously are urgently needed. It is known that CD4+ T cells are the main source of HIV-1 provirus DNA but defining a signatures of these cells is still challenging due to their scarcity in blood.

A considerable strength of this manuscript is the thorough validation of the platform. By using a model of in vitro infection of primary CD4+ T cells from an HIV-1 seronegative individual using a lab-adapted HIV with a known sequence, the authors elegantly identified epigenetic and cellular changes in HIV-1+ CD4+ T cells, which they used to build up a regression model that identifies with high accuracy several markers that predict infection. Another crucial part of the work that is extremely valuable was the use of autologous patient-derived HIV sequences, covering for the high mutational rate detected in the proviral sequence.

LIMITATIONS

1. Limited cell numbers (~150 HIV-1+ CD4+ T cells)

We thank the reviewer for this comment. We performed additional follow-up experiments with pre-ATI sampling of individuals A01 and A08 (similar to what was done for A09) to better assess the stability of the reservoir during ATI. Through these efforts, our combined dataset now has 213 HIV-1+ cells, reflecting a survey of 166,357 total cells and highlighting the inherent challenges in studying the HIV reservoir. The updated data is shown in Figures 3 and 4 and reaffirms many of our original findings.

With this new dataset:

- We added 14021 total cells from A01 pre-ATI and 18427 total cells from A08 pre-ATI. When combined with the original dataset, there are now 166,357 cells overall. Of these cells, 205 are HIV-infected and are annotated as T cells.
- Of the individuals with pre- and post-ATI timepoint analysis, we observed evidence of reservoir diversification post ATI, with a distinct appearance of virus within recently activated Tcm/Ttm populations. This is in line with the observed proviral diversification from previous studies (Bar et al, 2016 and Salantes et al. 2018) in the same individual after ATI, where new viral variants reseeded the reservoir during ATI. In contrast, our original observation of a stable reservoir composition in donor A09 before and after ATI corresponds to the individual in the trial with the lowest rebound viral load and also with VRC01-resistant provirus.
- The additional cells provided increased power in assessing surface antigen markers. CCR5, PD-1, and CD2 are upregulated in HIV+ T cells. Many of the markers identified have also been reported in prior literature (see original main text).
- Similar epigenetic signatures were detected that supported our manual annotation (note that number designations given to different modules have changed with this updated data). Tcm/Ttm cells showed greater enrichment of Tcf7 and Lef1 motifs (module 3) compared to Tem cells, while the Tem cells showed greater enrichment of Tbx21 (T-bet), Eomes, and interferon related motifs (modules 1, 6, and 7). MAIT cells, which can be heterogeneous in terms of different memory phenotypes, showed enrichment in motifs related to the nuclear factor-kappa B (NF-κB) transcription factor family including NFKB1, REL, RELA, and RELB (modules 3 and 4)
- Concordance with our original dataset where here we found BACH2 and AP-1 related family transcription factors such as Jun and Fos (Module 10) and certain NF-κB related motifs (Module 4) that showed relative enrichment in HIV+ cells compared to their respective phenotype-matched HIV- cells. Module 9 also displays enrichment, especially in the HIV+ Tcm cells compared to HIV- Tcm cells.

2. Requirement of the provirus to be in accessible chromatin regions, potentially biasing a towards a subset of latent reservoirs, albeit that most likely to contribute to ongoing clinical problems. Also, the methods could not differentiate between competent and defective viruses. Despite the technical limitations and the lack of identification of a "universal" signature of HIV-1+ cells, this manuscript provides a valuable technology, robust computational pipelines, and the creation of an atlas of HIV-1 infected cells that could be used and further curated by other HIV researchers, with the expectation in increasing the number of sequenced cells.

We agree that the main limitation of ASAPseq is with detection of infected cells being largely limited to cells with accessible provirus and have enhanced this point in our revised discussion. We are exploring novel single cell strategies that potentially could identify provirus in heterochromatin, but these assays are still in the conceptualization stage. As described in our original discussion, we believe that these cells may be a proxy for cells that are likely to reactivate during stimulation or latency reversal. Given the recent publications (Duette et al, 2022 and Imamichi et al, 2020) demonstrating that defective proviruses are still capable of producing proteins, cells with defective proviruses may therefore still be relevant in their ability to interface with the immune system and could potentially complicate latency reversal studies. We believe that our study is a foundational stepping stone for future studies to contribute to an "universal" signature given the immense heterogeneity found within infected cells.

SPECIFIC COMMENTS

1. HIV viral reservoirs are predominantly established in lymph nodes and the gut. Have the authors considered using LN samples to increase the number of cells harbouring HIV proviral DNA and ask whether or not the profiles of these cells are similar?

We share the same interest. We have profiled LN samples during chronic untreated infection and were able to detect relatively high levels of HIV-infected cells but we had initially excluded these data from the manuscript for scope and text limitations. After discussion with the editor, we have included these data in our revised manuscript.

Not surprisingly, we observed the Tfh cell in the LNs comprised the largest proportion of infected cells. Please refer to Supplemental Figures 6-9.

2. Three of the donors in the study previously experienced therapy interruption (enrolled in the VRC01 A5340 trial) but were under at least six months (for the post-ATI timepoint) of ART suppression at the time of analysis. It is unclear how long the subjects had been on ART before enrolment in the trial. Could the authors provide information on this and longitudinal viral load data, especially on the timepoint where ART was interrupted?

For all participants from the A5340 trial, the inclusion criteria included that all participants 1) had undetectable plasma viral load (< 50 copies/ml) for at least 24 weeks, 2) had no known treatment of ART during acute or early infection, and 3) did not have a NNRTI-based regimen. Longitudinal viral load data during ATI is documented in Bar et al, 2016 in Supplemental Figure S3. For time on ART in years prior to study entry, A01: 3.6 years, A08: 4.2 years, A09: 5.6 years. These values have been included in the revised Table 1.

3. The authors include an impressive array of 154 CITE-Seq antibodies for protein analysis. Can they provide the basis for their selection?

We chose this particular cocktail of antibodies because this was the largest commercially available panel and would therefore enable a more unbiased approach to assessing the surface antigen profiles. Each antibody in this pool was pre-titrated by BioLegend for use in sequencing-based methods. We will add more detail to the Methods section to explain our rationale for using this cocktail.

4. In Table 1, information on HIV clade and ethnicity would be valuable for any future cell atlas, as proposed by the authors in the discussion.

All donors were reported to be infected with HIV group M, subtype B viruses. Ethnicity was only reported in bulk in the original study and is available in their data (Table 1; Bar et al., 2016). We have included this in the main text.

5. It is striking that the authors found strong similarities in the phenotype of HIV-1+ CD4 T cell subsets before and after ART interruption. While possibly outside the scope of this manuscript, it would be interesting to dissect potential epigenetic changes that a temporal increase of viral load could have driven.

We agree with the reviewer that this would be an exciting line of inquiry. As described above, with the addition of two more longitudinal analyses, we are able to observe 'reseeding' of the reservoir into new subsets in one donor who experienced a higher level of viral rebound during ATI. However, there were insufficient infected cells from this single timepoint to develop an appropriately powered epigenetic analysis. To properly address this would require years to

perform and therefore move beyond what we could accomplish in a manuscript revision, but we agree that it is an important question.

6. Figure 4 Panels A-D have incomplete labeling of the x-axes. Panel B could be more precise by plotting the $-\text{Log}_{10}$ p-value. Finally, in panel F the y axis needs clarification.

We thank the reviewer for catching this. We have updated, corrected, and clarified the mentioned points.

Reviewer #2

The authors have developed a potentially important method for characterizing the HIV reservoir at the single cell level. They use a single cell ATAC-seq (scATACseq) approach to characterize individual cells that have reads containing HIV sequence. This is done first with cells infected in vitro with HIV and then with cells from persons living with HIV (PLWH) on antiretroviral therapy (ART). One goal seems to have been to identify surface markers unique to infected cells. The principal finding is that proviruses integrated into accessible chromosomal sites are present in a wide variety of different CD4 subpopulations.

The authors are to be congratulated for studying an important problem with a novel approach. This approach could make an important contribution to HIV cure research. Given the potential importance of the work, the authors should be encouraged to address the comments below to make the paper as strong as possible:

1) The paper is written in dense “genomics-speak” and is difficult to follow in places. For example, there is frequent use of the term “enrichment”, but the reader is left with little sense of the degree of enrichment from the statistical jargon that comprises much of the paper. Consider CD2 for example. This is a hit in some of the analyses but typically this protein shows a tight unimodal distribution on T cells. If there is biologically significant enrichment of markers like CD2 on infected cells, then it should be possible to test this experimentally by sorting CD2 high and low cells and measuring HIV.

We thank the reviewer for making this observation. The precision required of statistical language should not impede clarity. In a revised manuscript, we have contextualized the essential statistical points so that the biologic message is made clearer. We agree with the reviewer’s second point about sorting CD2+ cells, although this has been previously studied (Iglesias-Ussel et al, 2013). We have been tempted to sort cells by one marker or another to test for HIV DNA enrichment in our samples but have generally resisted this as we want the focus of this manuscript to be about the diversity of the cellular reservoir and the scope of analyses possible. We are less enthusiastic about using this small sample number to make a larger claim about other surface markers at this time. Still, it is reassuring that another marker, PD-1, was also identified as an enriched marker on HIV+ cells in our study given that PD-1-based sorting experiments have previously demonstrated increased HIV DNA (Perreau et al., 2013). We have clarified in the updated text for any markers that have been verified by previous enrichment studies.

2) Related to point 1, there is no attempt in the paper to experimentally validate any of the hits or relationships identified in the bioinformatic analysis.

Please see our above reply to Comment #1.

3) Some aspects of the method need to be clarified. The distribution of reads throughout the genome shows a peak in the middle of the genome that is not explained. The detection of HIV in B cells and monocytes (Fig.1b) casts some doubt on the reliability of the method.

The distribution of reads in the proviral genome is very fascinating and has been remarkably similar to a previous bulk ATACseq study on the proviral genome by Jefferys et al, 2021. While it is unclear the exact biological interpretation in either dataset, we posit that the resumption of reads after *pol* may be due to distinct nucleosomal structures or the presence of more stable protein binding at *pol*.

For the detection of HIV in B cells and monocytes, the detection of viral DNA in B cells and monocytes is not impossible given the documented phagocytic capabilities of B cells and monocytes as antigen processing cells. Furthermore, monocytes may also be infected as well. It is also plausible that these detections are multiplets or have insufficient data for correct calling. Regardless of their origin, we excluded these cells from our downstream analyses and focused only on cells found within T cell clusters.

We have included these points in the revised main text.

4) The study largely avoids key biological questions that the method could address, if used in combination with other assays. For example, what is the relationship between accessibility and viral gene expression?

We share a similar fascination with possible questions that could be addressed. A multiomic scATAC, scRNA, and scSAP approach (called TEAseq/DOGMAseq) is one of our future directions. Having the RNA component would allow us to explore if certain accessible proviruses are expressing viral transcripts under ART treatment and to seek the biological mechanisms behind selective viral expression. This work would take years and substantial financial resources and is therefore not possible for a revised manuscript. However, given its importance to the science and to our own current endeavors, we have included mention of this important point in the discussion.

5) It is unclear to what extent the findings contradict the large number of papers that claim that HIV is preferentially located in particular subsets? A better effort could be made to cite these papers and also papers that show wide distribution of HIV in different subsets.

We agree with the reviewer that comments comparing and contrasting to prior manuscripts are lacking. Our revised manuscript has included more references to prior literature. However, given our discussion in points 1 and 2 above, we would prefer not to make too large a claim about contrasting data given that single cell methods necessarily have a lower individual n than studies that sort for PCR-based measurement assays from multiple participants.

6) It is likely that many of the proviruses detected in the assay are defective. This issue is only briefly mentioned, and it is unclear how this would affect the conclusions.

We thank the reviewer for this comment and reference our previous reply to Reviewer #1. While we are able to infer viral intactness based on read recovery, this does not provide a definitive

assessment of viral sequence. Based upon this metric, indeed many of the proviruses are likely defective. We have elaborated on these points in our discussion.

7) Similarly, it is likely that many of the proviruses detected are from expanded clones. It would be helpful to know how members of the same clone of HIV infected CD4+ T cells differ with regard to features measured in this assay.

We agree that many proviruses are likely to be from expanded clones. However, it is not possible in this current assay to assess clonality as the proviral coverage has been shown to be insufficient for calling clonality (Patro et al., 2019). Inferring TCR sequence by ATAC to determine T cell clonality is dependent on tagmentation and sequencing coverage of the TCR and is currently technically infeasible. A 5' scRNAseq component is the current standard to assess clonality in a single-cell multiomic assay, but a multiomic 5' scRNA + scATAC assay is not available commercially. We have elaborated on these points in our discussion.

8) The legends for the main and supplemental figures should be rewritten to clearly state what was tested in each panel and what the controls were.

We have updated the legends to more clearly explain what was tested and the appropriate controls if applicable.

Decision Letter, first revision:

Our ref: NI-RS34295A

10th Oct 2022

Dear Dr. Betts,

Thank you for submitting your revised manuscript "Profound phenotypic and epigenetic heterogeneity of the HIV-1 infected CD4+ T cell reservoir" (NI-RS34295A). It has now been seen by the original referees and their comments are below. The reviewer finds that the paper has improved in revision, and therefore we'll be happy in principle to publish it in Nature Immunology, pending minor revisions to comply with our editorial and formatting guidelines.

We will now perform detailed checks on your paper and will send you a checklist detailing our editorial and formatting requirements in about a week. Please do not upload the final materials and make any revisions until you receive this additional information from us.

If you had not uploaded a Word file for the current version of the manuscript, we will need one before beginning the editing process; please email that to immunology@us.nature.com at your earliest convenience.

Thank you again for your interest in Nature Immunology Please do not hesitate to contact me if you have any questions.

Sincerely,

Jamie D.K. Wilson, D.Phil
Chief Editor
Nature Immunology
212 726 9207
j.wilson@us.nature.com

Reviewer #2 (Remarks to the Author):

The authors have discussed all of the concerns raised in the review. There is little in the way of new data, but significant clarification of the text and discussion of the limitations of the approach.

Final Decision Letter:

Subject: Decision on Nature Immunology submission NI-RS34295B

Message: In reply please quote: NI-RS34295B

Dear Dr. Betts,

I am delighted to accept your manuscript entitled "Profound phenotypic and epigenetic heterogeneity of the HIV-1 infected CD4+ T cell reservoir" for publication in an upcoming issue of Nature Immunology.

Over the next few weeks, your paper will be copyedited to ensure that it conforms to Nature Immunology style. Once your paper is typeset, you will receive an email with a link to choose the appropriate publishing options for your paper and our Author Services team will be in touch regarding any additional information that may be required.

Please note that *Nature Immunology* is a Transformative Journal (TJ). Authors may publish their research with us through the traditional subscription access route or make their paper immediately open access through payment of an article-processing charge (APC). Authors will not be required to make a final decision about access to their article until it has been accepted. [Find out more about Transformative Journals](https://www.springernature.com/gp/open-research/transformative-journals).

Authors may need to take specific actions to achieve [compliance with funder and institutional open access mandates](https://www.springernature.com/gp/open-research/funding/policy-compliance-faqs). If your research is supported by a funder that requires immediate open access (e.g. according to [Plan S principles](https://www.springernature.com/gp/open-research/plan-s-compliance)) then you should select the gold OA route, and we will direct you to the compliant route where possible. For authors selecting the subscription publication route, the journal's standard licensing terms will need to be accepted, including [self-archiving policies](https://www.springernature.com/gp/open-research/policies/journal-policies). Those licensing terms will supersede any other terms that the author or any third party may assert apply to any version of the manuscript.

Your paper will be published online soon after we receive your corrections and will appear

in print in the next available issue. Content is published online weekly on Mondays and Thursdays, and the embargo is set at 16:00 London time (GMT)/11:00 am US Eastern time (EST) on the day of publication. Now is the time to inform your Public Relations or Press Office about your paper, as they might be interested in promoting its publication. This will allow them time to prepare an accurate and satisfactory press release. Include your manuscript tracking number (NI-RS34295B) and the name of the journal, which they will need when they contact our office.

About one week before your paper is published online, we shall be distributing a press release to news organizations worldwide, which may very well include details of your work. We are happy for your institution or funding agency to prepare its own press release, but it must mention the embargo date and Nature Immunology. Our Press Office will contact you closer to the time of publication, but if you or your Press Office have any enquiries in the meantime, please contact press@nature.com.

Also, if you have any spectacular or outstanding figures or graphics associated with your manuscript - though not necessarily included with your submission - we'd be delighted to consider them as candidates for our cover. Simply send an electronic version (accompanied by a hard copy) to us with a possible cover caption enclosed.

Please note that we encourage the authors to self-archive their manuscript (the accepted version before copy editing) in their institutional repository, and in their funders' archives, six months after publication. Nature Portfolio recognizes the efforts of funding bodies to increase access of the research they fund, and strongly encourages authors to participate in such efforts. For information about our editorial policy, including license agreement and author copyright, please visit www.nature.com/ni/about/ed_policies/index.html

Sincerely,

Jamie D.K. Wilson, D.Phil
Chief Editor
Nature Immunology
212 726 9207
j.wilson@us.nature.com